# Dynamic diversity of SARS-CoV-2 genetic mutations in a lung transplantation patient with persistent COVID-19

Hidetoshi Igari [1,2,3,4,11] ✉, Seiichiro Sakao [5,6], Takayuki Ishige [7,11] ✉, Kengo Saito[8], Shota Murata [7], Misuzu Yahaba[1], Toshibumi Taniguchi [1,4], Akiko Suganami[9], Kazuyuki Matsushita [7], Yutaka Tamura[9], Takuji Suzuki[2,5,10] & Eiji Ido [1,8,11] ✉

Numerous SARS-CoV-2 variant strains with altered characteristics have emerged since the onset of the COVID-19 pandemic. Remdesivir (RDV), a ribonucleotide analogue inhibitor of viral RNA polymerase, has become a valuable therapeutic agent. However, immunosuppressed hosts may respond inadequately to RDV and develop chronic persistent infections. A patient with respiratory failure caused by interstitial pneumonia, who had undergone transplantation of the left lung, developed COVID-19 caused by Omicron BA.5 strain with persistent chronic viral shedding, showing viral fusogenicity. Genome-wide sequencing analyses revealed the occurrence of several viral mutations after RDV treatment, followed by dynamic changes in the viral populations. The C799F mutation in nsp12 was found to play a pivotal role in conferring RDV resistance, preventing RDV-triphosphate from entering the active site of RNA-dependent RNA polymerase. The occurrence of diverse mutations is a characteristic of SARS-CoV-2, which mutates frequently. Herein, we describe the clinical case of an immunosuppressed host in whom inadequate treatment resulted in highly diverse SARS-CoV-2 mutations that threatened the patient's health due to the development of drug-resistant variants.

Numerous SARS-CoV-2 variants with altered infectivity, pathogenicity, antiviral drug resistance, and ability to escape host immunity, have emerged during the COVID-19 pandemic. Remdesivir (RDV) is a direct-acting nucleotide prodrug inhibitor of SARS-CoV-2 RNA-dependent RNA polymerase (nsp12, RdRp)[1]. RDV improves clinical outcomes in patients hospitalized for moderate to severe COVID-19[2]. However, immunosuppressed hosts such as organ transplant recipients who are infected with SARS-CoV-2, may not respond adequately to RDV treatment, and develop chronic persistent infections[3–7]. Both cell-mediated immunity and humoral immunity are required to control SARS-CoV-2 infection[3,8,9]. Furthermore, inadequate RDV treatment may result in genetic mutations, leading to the emergence of novel SARS-CoV-2 variants[10].

[1]Department of Infectious Diseases, Chiba University Hospital, Chiba, Chiba, Japan. [2]Future Mucosal Vaccine Research and Development Center, Chiba University Hospital, Chiba, Chiba, Japan. [3]COVID-19 Vaccine Center, Chiba University Hospital, Chiba, Chiba, Japan. [4]Research Institute of Disaster Medicine, Chiba University, Chiba, Chiba, Japan. [5]Department of Respiratory Medicine, Chiba University Hospital, Chiba, Chiba, Japan. [6]Department of Pulmonary Medicine, School of Medicine, International University of Health and Welfare, Narita, Chiba, Japan. [7]Division of Laboratory Medicine, Chiba University Hospital, Chiba, Chiba, Japan. [8]Department of Molecular Virology, Graduate School of Medicine, Chiba University, Chiba, Chiba, Japan. [9]Department of Bioinformatics, Graduate School of Medicine, Chiba University, Chiba, Chiba, Japan. [10]Synergy Institute for Futuristic Mucosal Vaccine Research and Development, Chiba University, Chiba, Chiba, Japan. [11]These authors contributed equally: Hidetoshi Igari, Takayuki Ishige, Eiji Ido. ✉e-mail: igari_h@chiba-u.jp; ishige-t@chiba-u.jp; idoe@chiba-u.jp

Herein, we report the case of a patient who had undergone transplantation of the left lung and had since been treated with immunosuppressive drugs. After SARS-CoV-2 infection, the patient developed a chronic persistent infection despite treatment with RDV. We are particularly interested in what kind of events associated with pathogenicity and drug resistance have occurred in this process. During the lingering course of infection, SARS-CoV-2 acquired dynamic and diverse mutations across its genome. In this study, we show that some of the mutations including C799F in RdRp have triggered to generate diversified SARS-CoV-2 variants, leading to various phenotypes such as increased cytopathogenicity and drug resistance.

## Results

### Clinical course and laboratory data

The patient was at the age of 50s, diagnosed with interstitial pneumonia (IP) caused by rheumatoid arthritis that had developed 17 years earlier. The patient developed respiratory failure due to IP. Three years earlier, this person underwent transplantation of the left lung from a brain-dead person. Since then, the patient has been treated with immunosuppressive drugs such as tacrolimus (1.5 mg/day), mycophenolate mofetil (500 mg/day), and prednisolone (8 mg/day) (Fig. 1). Itraconazole, trimethoprim/sulfamethoxazole, and valganciclovir were prescribed for the prevention of opportunistic infections due to immunosuppressive drug use after lung transplantation. The patient was also periodically administered immunoglobulin. The patient presented with a sore throat and fever that had appeared the day before admission. The qualitative nasopharyngeal swab test result was positive for the SARS-CoV-2 antigen. This person was then admitted to our hospital (Day 1). The patient had received three doses of SARS-CoV-2

mRNA vaccine, with the final vaccination taking place 4 months before COVID-19 onset. Tixagevimab/cilgavimab was injected intramuscularly 8 months before onset. The patient's immunoglobulin G (IgG) titre against the SARS-CoV-2 spike (S) protein was elevated, while being very low compared to that after mRNA vaccination and SARS-CoV-2 infection. IgG against the nucleocapsid (N) protein was negative (Table 1).

The clinical course and laboratory data of this case are summarised in Fig. 1 and Table 1. After confirming the improvements of symptoms by way of the first course of RDV medication, the patient was discharged on Day 11. This person was re-admitted to the hospital on Day 22 with pneumonia caused by COVID-19. The patient also developed pneumonia caused by *Escherichia coli* infection. Therefore, a second course of RDV and antimicrobial (cefepime) treatment was initiated. From Day 28, respiratory and renal failure progressed. On Day 36, the patient was transferred to the ICU to receive haemodialysis and high-flow oxygen therapy. During this time, the patient developed invasive pulmonary aspergillosis caused by *Aspergillus tubingensis*, and antifungal treatment with voriconazole and micafungin was initiated. On Day 44, neutrophil counts decreased down to 62/μl. Valganciclovir was suspected to cause the neutropenia and was therefore discontinued, whereafter a filgrastim biosimilar (G-CSF: granulocyte colony-stimulating factor) was started. Since *Pneumocystis* pneumonia was also suspected, the dose of trimethoprim/sulfamethoxazole was increased from prophylactic to therapeutic.

On Day 62, the patient developed a persistent fever and received a third course of RDV was initiated. Subsequently, the C-reactive protein levels improved, and the RT-PCR cycle threshold (Ct) value indicative of SARS-CoV-2 levels from nasopharyngeal swab samples increased. However, the patient continuously required haemodialysis for chronic

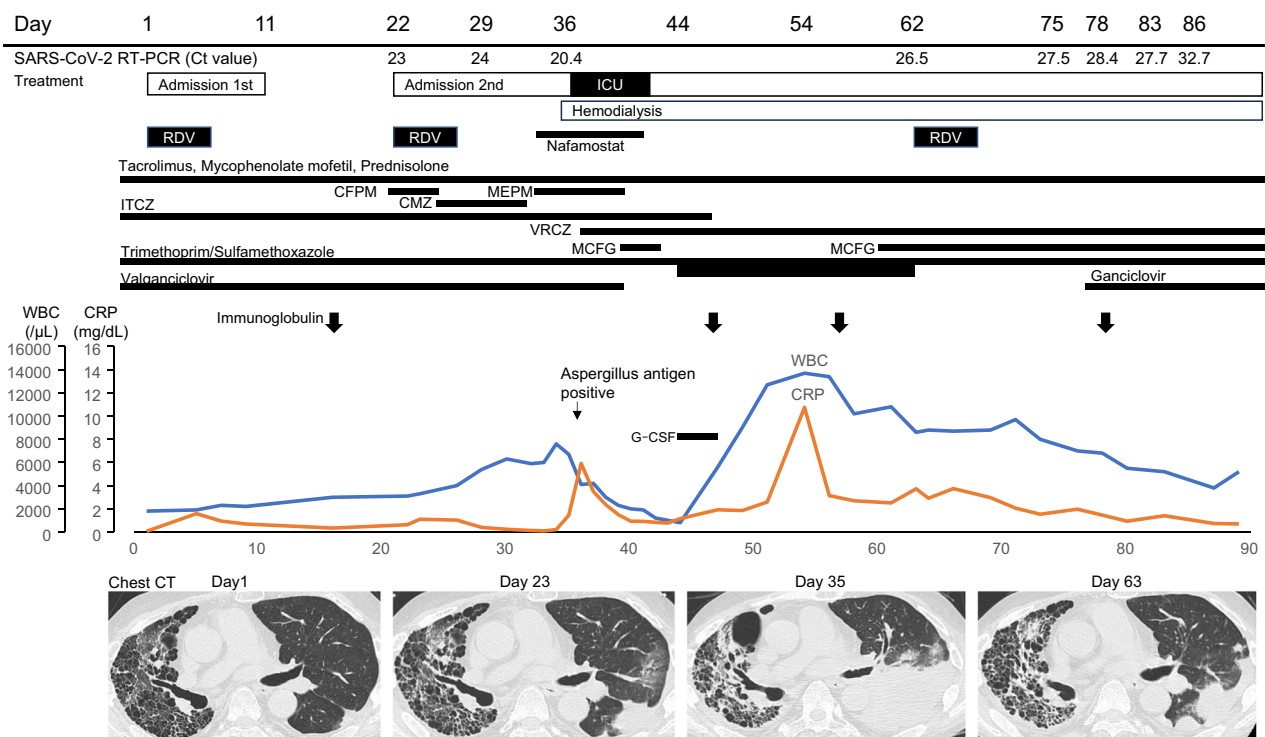

**Fig. 1 | Clinical course.** After the first course of RDV, the patient's symptoms improved (Day 11). The patient was re-admitted to the hospital on Day 22 due to worsening physical symptoms. From Day 28, respiratory and renal failure progressed. On Day 36, the patient was transferred to the ICU to undergo haemodialysis. During this time, the patient developed invasive pulmonary aspergillosis caused by *Aspergillus tubingensis*, and antifungal treatment with voriconazole and micafungin was initiated. On Day 44, neutrophils decreased down to 62/μl. Valganciclovir was the suspected drug causing neutropenia and was discontinued.

Treatment with a filgrastim biosimilar (G-CSF: granulocyte colony-stimulating factor) was initiated. Since *Pneumocystis* pneumonia was also suspected, the dose of trimethoprim/sulfamethoxazole was increased from prophylactic to therapeutic. On Day 62, the patient had a persistent fever; the Ct value of nasopharyngeal specimen was as low as 26.5. Therefore, the third course of RDV was initiated. RDV remdesivir, Ct Cycle threshold, CFPM cefepime, CMZ cefmetazole, MEPM meropenem, ITCZ itraconazole, VRCZ voriconazole, MCFG micafungin, G-CSF granulocyte colony-stimulating factor.

**Table 1 | Clinical course and laboratory data of the patient**

|  | Day 1 | Day 22 | Day 29 | Day 36 | Day 44 | Day 54 | Day 62 | Day 75 | Day 78 | Day 83 | Day 86 |
|---|---|---|---|---|---|---|---|---|---|---|---|
| Oxygen (l/min) | Room air | 3 | 1 | HFNC: 60 l/min 60% | 2 | 1 | 1 | 1 | 1 | 1 | 1 |
| SpO₂ (%) | 93 | 94 | 98 | 100 | 99 | 97 | 96 | 97 | 97 | 97 | 98 |
| Body temperature (°C) | 38.5 | 38.9 | 37.1 | 38.8 | 37.0 | 37.1 | 36.5 | 37.1 | 36.9 | 36.5 | 36.9 |
| CRP (mg/dl) | 0.06 | 0.63 | 0.23 | 5.91 | 1.16 | 10.75 | 3.72 | 1.97 | 1.46 | 1.4 | 0.74 |
| WBC (/µl) | 1800 | 1800 | 6300 | 4100 | 800 | 13,700 | 8600 | 7000 | 6800 | 5200 | 3800 |
| Neutrophil (/µl) | NA | NA | 6048 | 3813 | 62 | 1178 | 7568 | 5460 | 5168 | 3640 | NA |
| Lymphocyte (/µl) | NA | NA | 189 | 0 | 7 | 82 | 430 | 700 | 748 | 832 | NA |
| Platelets (/µl)(×1000) | 114 | 142 | 161 | 51 | 42 | 201 | 227 | 134 | 125 | 136 | 102 |
| RT-PCR (Ct) | NA | 23 | 24 | 20.4 | NA | NA | 26.5 | 27.5 | 28.4 | 27.7 | 32.7 |
| Anti-SpikeIgG (U/ml) | NA | NA | NA | NA | NA | NA | NA | 190 | 165 | 153 | 142 |
| Anti-nucleocapsid IgG | NA | NA | NA | NA | NA | NA | NA | Negative | Negative | Negative | Negative |

*HFNC* high-flow nasal cannula, *CRP* C-reactive protein, *WBC* white blood cell, *NA* not applicable, *Ct* cycle threshold.

renal failure and oxygen therapy for chronic respiratory failure. The patient died 7 months after infection.

## SARS-CoV-2 genetic mutations
Whole-genome sequencing revealed several amino acid mutations, especially in the S protein and ORF 1ab. The average read depth of each sample was as follows: Day 22, 899; Day 29, 998; Day 36, 791; Day 62, 664; Day 75, 609; Day 78, 788; Day 83, 1062; and Day 86, 861. The amino acid mutations compared with those in the index strain of SARS-CoV-2 (NC_045512.2, Pango lineage B) were categorised as major and minor (by a frequency of more and less than 50%, respectively), and all mutations with a frequency of 5% or higher are shown in Tables 2 and 3 and Supplementary Table 1. Fifty-four major mutations were identified on Day 22, and the number increased to 70 by Day 86. Mutations in the S protein, identified at 39 loci, were the most common, followed by those in ORF1ab at 20 (ORF1a and ORF1b were 11 and nine, respectively) locations and N protein mutations at six loci (Table 2). The number of minor mutations reached a maximum of 64 on Day 36 and decreased thereafter. ORF1ab harboured the highest number of mutations, identified at 50 (ORF1a and ORF1b harboured 21 and 29, respectively) loci, followed by the S protein, with mutations in seven loci. Notably, a peak in minor mutations was first observed, followed by an increase in the number of major mutations (Supplementary Fig. 1).

## Isolation of various SARS-CoV-2 mutants
Viral isolation was attempted using Vero/TMPRSS2 cells, a SARS-CoV-2-susceptive cell line[11], cultured in 6-well-plates with nasopharyngeal specimens collected after the patient was re-admitted. Typical virus-causing cytopathogenic effects (CPE, usually rounded cells) appeared 2–4 days post-infection (d.p.i.). On Day 22, an Omicron BA.5 variant, termed CH-LT1, and a V792I mutant in nsp12, termed CH-LT1m, were isolated. On Day 29, a C799F mutant in nsp12, termed CH-LT2m, and on Day 36, a M794I mutant, termed CH-LT3m, were identified. The viral isolate obtained on Day 62 (CH-LT4) had considerably diversified mutations, including C799F in nsp12 and others in the S protein (Table 3, Fig. 2 and Supplementary Table 1). Virus isolates were also obtained at later days (CH-LT6 on Day 78 and CH-LT7 on Day 83). The virus was not isolated from the specimens on Day 75 and Day 86. The sequences of these later isolates were confirmed via RT-PCR followed by conventional sequencing and found to be identical to the consensus sequences obtained through whole-genome sequencing in the nsp12 region and the S-protein gene, the only regions exhibiting genetic drift. Viral shedding was confirmed on the first day of re-admission (Day 22) and thereafter.

The Nextclade-based phylogenetic placement of SARS-CoV-2 strains is shown in Fig. 3a, all belonging to BA.5.2.1. Figure 3b displays the phylogenetic tree with the entire genome sequences of current strains along with a representative variant of concern (VOC). Notably, the SARS-CoV-2 strains from the current case exhibited a range of diverse and dynamic mutations.

## Switch to fusogenic strains
CPEs caused by the respective isolates in VeroE6/TMPRSS2 cells at three d.p.i. are shown in Fig. 4. Omicron BA.5 strains cause cell rounding as a typical CPE, with occasional ballooning. The isolates obtained on earlier days (from CH-LT1 to CH-LT3m) caused similar CPEs (Fig. 4a–d). Remarkably, the viral properties of CH-LT4 obtained on Day 62 appeared to change drastically and become more fusogenic (Fig. 4e). In addition to cell-rounding, large clumps of cells that merged with the adjacent cells were frequently observed. This fusogenic nature was maintained in later isolates (CH-LT6 and CH-LT7, Days 78 and 83, respectively) (Fig. 4f, g). We confirmed this switch to fusogenic strains by three independent experiments.

## Growth kinetics of the viral isolates
The growth kinetics of all isolates obtained in the present clinical course are shown in Fig. 5. The experiments were performed in triplicate. The earlier isolates (CH-LT1 to CH-LT3m) propagated very fast and reached a peak within 2–3 d.p.i. whereas the growth of later isolates (CH-LT4 to CH-LT7) was delayed 1–2 days, reaching a peak at 4–5 d.p.i. even if we adjusted the same MOI (0.01) for all isolates. In addition to acquiring fusogenicity, the later isolates exhibited altered growth speed. Nonetheless, it should be noted that the final progeny production levels were almost the same among all isolates.

## Acquisition of RDV resistance via nsp12 mutations
To determine whether the isolates acquired any resistance against RDV via mutations in nsp12, we performed RDV resistance assays (Fig. 6)[12]. CH-LT1, an Omicron BA.5 strain, could not replicate in the presence RDV at 5 µM or higher concentrations (median effect concentration, $EC_{50} = 0.8$ µM, 95% confidence interval (CI): 0.7–1.0 µM). Meanwhile, CH-LT1m, CH-LT2m, and CH-LT3m, harbouring the V792I (G15814A), C799F (G15836T), and M794I (G15822T) mutations in nsp12, respectively, acquired some resistance against RDV. The $EC_{50}$ of CH-LT1m, CH-LT2m, and CH-LT3m was 1.4 (95% CI: 1.1–1.7 µM), 2.7 (95% CI: 2.0–3.3 µM), and 1.9 µM (95% CI: 1.4–2.5 µM), respectively. Obviously, the higher concentration of RDV were necessary to sufficiently inhibit the viral growth of CH-LT1m, CH-LT2m and CH-LT3m than in the case of CH-LT1. Among the three isolates, the most evident resistance was exhibited by CH-LT2m harbouring the C799F mutation, which could

**Table 2 | Number of amino acid mutations in different gene regions**

| Gene | | Group[a] | Day 22 | Day 29 | Day 36 | Day 62 | Day 75 | Day 78 | Day 83 | Day 86 | cf BA.5.2.1 (Outbreak.info) |
|---|---|---|---|---|---|---|---|---|---|---|---|
| ORF1a | | Major | 9 | 9 | 9 | 10 | 10 | 10 | 10 | 11 | 8 |
| | | Minor | 9 | 4 | 21 | 11 | 10 | 9 | 14 | 7 | 0 |
| ORF1b | nsp12 is encoded | Major | 4 | 5 | 5 | 7 | 9 | 9 | 7 | 9 | 4 |
| | | Minor | 5 | 3 | 29 | 11 | 3 | 3 | 8 | 3 | 0 |
| ORF2 | Spike protein | Major | 31 | 31 | 31 | 35 | 38 | 38 | 37 | 39 | 31 |
| | | Minor | 0 | 1 | 7 | 13 | 3 | 7 | 10 | 6 | 0 |
| ORF3a | | Major | 1 | 1 | 1 | 1 | 1 | 1 | 1 | 1 | 1 |
| | | Minor | 0 | 1 | 2 | 0 | 0 | 0 | 1 | 0 | 0 |
| ORF4 | Envelope protein | Major | 1 | 1 | 1 | 1 | 1 | 1 | 1 | 1 | 1 |
| | | Minor | 0 | 0 | 0 | 0 | 0 | 0 | 0 | 0 | 0 |
| ORF5 | Membrane protein | Major | 3 | 3 | 3 | 3 | 3 | 3 | 3 | 3 | 3 |
| | | Minor | 2 | 0 | 2 | 0 | 0 | 0 | 0 | 0 | 0 |
| ORF6 | | Major | 0 | 0 | 0 | 0 | 0 | 0 | 0 | 0 | 0 |
| | | Minor | 2 | 3 | 2 | 0 | 0 | 2 | 1 | 0 | 0 |
| ORF7a/b | | Major | 0 | 0 | 0 | 0 | 0 | 0 | 0 | 0 | 0 |
| | | Minor | 0 | 1 | 0 | 0 | 0 | 0 | 0 | 0 | 0 |
| ORF8 | | Major | 0 | 0 | 0 | 0 | 1 | 1 | 1 | 0 | 0 |
| | | Minor | 0 | 0 | 0 | 0 | 0 | 0 | 1 | 0 | 0 |
| ORF9 | Nucleocapsid protein | Major | 5 | 5 | 5 | 5 | 5 | 5 | 5 | 6 | 4 |
| | | Minor | 1 | 1 | 1 | 3 | 2 | 3 | 2 | 2 | 0 |
| ORF10 | | Major | 0 | 0 | 0 | 0 | 0 | 0 | 0 | 0 | 0 |
| | | Minor | 0 | 0 | 0 | 0 | 0 | 0 | 0 | 0 | 0 |
| All | | Major | 54 | 55 | 55 | 62 | 68 | 68 | 65 | 70 | 52 |
| | | Minor | 19 | 14 | 64 | 38 | 18 | 24 | 37 | 18 | 0 |
| | | Total | 73 | 69 | 119 | 100 | 86 | 92 | 102 | 88 | 52 |

[a]The major and minor mutations in this table are defined by a frequency of more and less than 50%, respectively.

replicate to a detectable level even in the presence of RDV at 5 μM or higher concentrations. We did not include the later isolates (CH-LT4, CH-LT6, and CH-LT7) in this RDV resistance assay. Since the viral properties of these greatly changed with regard to both growth kinetics and viral fusogenicity, it was difficult to apply the unified experimental condition for assessing RDV resistance. Thus, we focused on the role of mutations which occurred in nsp12.

We separately performed comparison experiments for the later isolates with the original CH-LT1, in the presence or absence of 5 μM RDV at 3 and 4 d.p.i. As shown in Supplementary Fig. 2, CH-LT4 and later isolates could propagate at least to some extent (approximately at least one to two orders of magnitude lower) in the presence of 5 μM RDV, whereas CH-LT1 was completely suppressed to undetectable levels. These results indicate that the later isolates continuously maintained the properties of RDV resistance.

### in silico simulation of SARS-CoV-2 RdRp

To interpret the acquisition of RDV resistance, we performed in silico simulation of the SARS-CoV-2 RdRp with RDV-triphosphate (RDV-TP). We constructed the three-dimensional structure of SARS-CoV-2 RdRp with V792I, M794I, and C799F mutations and found that these mutations created two hydrophobic clusters, designated Site I and Site II, in Domain-II of the SARS-CoV-2 RdRp (Fig. 7). Site I creates a large hydrophobic cluster by adding C799F to join hydrophobic amino acid side chains of I779 and W800, and Site II extends the hydrophobic cluster via V792I and M794I, flanking the conserved hydrophobic residue F793 (Fig. 7a). In addition, Sites I and II are located close to K798 presumed to play a pivotal role in the active site-linked polymerase activity of SARS-CoV-2 RdRp[13,14]. To analyze the contribution of Sites I and II to RDV resistance, we employed in silico molecular docking simulation. As shown in the

expansion area in Fig. 7b, c, the results of molecular docking simulation suggested that RDV-TP was trapped through hydrophobic interactions with I779 (Site I) and F793 (Site II), preventing entry to the active site of SARS-CoV-2 RdRp.

### Discussion

Previous reports have described cases of SARS-CoV-2 infection in immunosuppressed patients, which led to chronic persistent infection due to an inadequate response to RDV[3–7]. There is concern that viral evolution may accelerate, leading to the emergence of new, genetically diverse variants[3]. SARS-CoV-2 variants can affect the transmission, cytopathogenicity, drug resistance, as well as responses to vaccine-induced immunity and antibody treatment. Our patient was infected with SARS-CoV-2 during immunosuppressive therapy following lung transplantation. Despite difficulties in the clinical course, we were able to use virus isolates to analyze dynamic changes in amino acid mutations, resistance to RDV, and CPE.

There have been reports of amino acid mutations during similar clinical courses[5,12]. In the current study, all amino acid mutations throughout the viral genome were analyzed. When limited to major mutations with a frequency of 50% or higher, there were 54 mutations at Day 22, and by Days 62 and 86, increasing to 62 and 70, respectively (Table 2). Although this observation was made in an immunosuppressed host, to the best of our knowledge, this is the highest number of amino acid mutations reported in a single clinical case.

Furthermore, it should be noted that minor mutations with a frequency lower than 50% were observed to increase within 2–3 weeks of the start of both the second and third RDV administration (Table 2 and Supplementary Fig. 1). Most of the minor mutations were thereafter eliminated, but some, such as C799F, persisted to become major mutations. Under a series of processes establishing selective pressure,

**Table 3 | Frequency of genetic and amino acid mutations in SARS-CoV-2**

| Nucleotide mutation | Amino acid mutation | Day 22 | Day 29 | Day 36 | Day 62 | Day 75 | Day 78 | Day 83 | Day 86 |
|---|---|---|---|---|---|---|---|---|---|
| T13924G | ORF1b:W153G(nsp12:W162G) | NA | NA | 0.09 | 0.04 | NA | NA | NA | NA |
| A13928T | ORF1b:Y154F (nsp12:Y163F) | NA | NA | 0.11 | 0.05 | NA | NA | NA | NA |
| A14359C | ORF1b:I298L (nsp12:I307L) | NA | NA | NA | NA | 0.50 | 0.59 | NA | NA |
| G14831A | ORF1b:C455Y (nsp12:C464Y) | 0.05 | 0.03 | NA | NA | NA | NA | NA | NA |
| G14831T | ORF1b:C455F (nsp12:C464F) | 0.05 | NA | NA | NA | NA | NA | NA | NA |
| T14996C | ORF1b:M510T (nsp12:M519T) | NA | NA | 0.06 | NA | NA | NA | NA | NA |
| G15451A | ORF1b:G662S (nsp12:G671S) | NA | NA | NA | 0.92 | 1 | 1 | 1 | 1 |
| T15682A | ORF1b:Y739N (nsp12:Y748N) | NA | NA | 0.04 | 0.05 | NA | NA | NA | 0.04 |
| T15685A | ORF1b:L740M (nsp12:L749M) | NA | NA | 0.05 | 0.06 | NA | 0.03 | 0.03 | 0.03 |
| G15769T | ORF1b:A768S (nsp12:A777S) | 0.16 | NA | NA | NA | NA | NA | NA | NA |
| G15814A | ORF1b:V783I (nsp12:V792I) | 0.26 | 0.31 | NA | NA | NA | NA | NA | NA |
| G15822A | ORF1b:M785I (nsp12:M794I) | NA | NA | 0.06 | 0.04 | NA | NA | NA | NA |
| G15822T | ORF1b:M785I (nsp12:M794I) | NA | NA | 0.51 | NA | NA | NA | NA | NA |
| T15835C | ORF1b:C790R (nsp12:C799R) | NA | 0.05 | 0.05 | NA | NA | NA | NA | NA |
| G15836T | ORF1b:C790F (nsp12:C799F) | NA | 0.58 | 0.14 | 0.96 | 1 | 1 | 1 | 1 |
| T15877C | ORF1b:C804R (nsp12:C813R) | NA | NA | 0.05 | NA | NA | NA | NA | NA |
| T21656C | S:F32L | NA | NA | NA | 0.18 | NA | NA | NA | NA |
| G21800T | S:D80Y | NA | NA | NA | NA | NA | NA | NA | 0.06 |
| G21830T | S:V90F | NA | NA | 0.05 | NA | NA | NA | NA | NA |
| 21992_21994del | S:Y144del | NA | NA | NA | 0.82 | 0.78 | 0.79 | 0.83 | 0.87 |
| T22092A | S:M177K | NA | NA | NA | NA | NA | 0.04 | 0.27 | 0.80 |
| G22100A | S:E180K | NA | NA | NA | NA | 0.06 | 0.08 | NA | NA |
| G22103C | S:G181R | NA | NA | NA | NA | NA | 0.08 | NA | 0.18 |
| G22103A | S:G181R | NA | NA | NA | NA | 0.52 | 0.65 | 0.07 | NA |
| G22203T | S:R214L | NA | NA | NA | 0.93 | 1 | 1 | 1 | 1 |
| C22448T | S:L296F | NA | NA | NA | NA | NA | NA | 0.05 | NA |
| C22550T | S:P330S | NA | NA | 0.14 | NA | NA | NA | NA | NA |
| G22599C | S:R346T | NA | NA | NA | 1 | 1 | 1 | 1 | 1 |
| T22667C | S:Y369H | NA | NA | 0.10 | NA | NA | NA | NA | NA |
| A22728G | S:D389G | NA | NA | NA | 0.49 | 0.85 | 0.97 | 0.73 | 0.16 |
| T23018G | S:F486V | 1 | 1 | 0.98 | 0.05 | NA | NA | NA | NA |
| G23072A | S:G504S | NA | NA | 0.06 | NA | NA | NA | NA | NA |
| G23073A | S:G504D | NA | 0.31 | NA | NA | NA | NA | NA | NA |
| C23202A | S:T547K | NA | NA | NA | NA | 0.15 | NA | 0.26 | 0.82 |
| G23405T | S:V615F | NA | NA | NA | NA | NA | NA | 0.07 | NA |
| G23463T | S:R634L | NA | NA | NA | 0.23 | 0.16 | NA | 0.27 | 0.83 |
| G23463A | S:R634H | NA | NA | NA | 0.10 | NA | NA | NA | NA |
| T23487G | S:V642G | NA | NA | NA | 0.46 | 0.82 | 0.97 | 0.74 | 0.17 |
| C23589A | S:T676N | NA | NA | NA | 0.92 | 1 | 1 | 1 | 1 |
| A23592C | S:Q677P | NA | NA | NA | 0.08 | NA | NA | NA | NA |
| A23598G | S:N679S | NA | NA | NA | NA | NA | NA | NA | 0.16 |
| C24117T | S:A852V | NA | NA | NA | 0.08 | NA | NA | 0.28 | 0.90 |
| G25145A | S:E1195K | NA | NA | NA | 0.94 | 1 | 1 | 1 | 1 |
| A25325T | S:K1255* | NA | NA | 0.06 | 0.05 | NA | NA | NA | NA |
| G25352C | S:V1264L | NA | NA | NA | NA | NA | NA | 0.05 | NA |

In this table, mutations in ORF1b (nsp12) and S genes with a frequency of 5% or higher, as determined using whole-genome sequencing, are listed.
*NA* not applicable.

the virus may have characteristics different from those of conventional Omicron strains.

During the second course of RDV treatment, we identified three amino acid mutations in nsp12, i.e., V792I, C799F, and M794I. V792I was identified on Days 22 and 29, with frequencies of 26% and 31%, respectively (Table 3). C799F was first identified on Day 29, with a frequency of 58%. M794I was identified only on Day 36, with a frequency of 51%, and was never detected later. An in vitro study has

shown that V166A, N198S, S759A, V792I, and C799F/R, which are present in RdRp, are amino acid mutations leading to RDV resistance[12]. However, clinical reports on the matter are limited. A case report described a renal transplant patient with COVID-19 who developed an amino acid mutation in V792I after RDV treatment[6]. Here, V792I was identified on Day 22 prior to the commencement of the second RDV treatment. It is therefore highly plausible that V792I may have emerged from the first course of RDV treatment, whereas C799F and M794I may

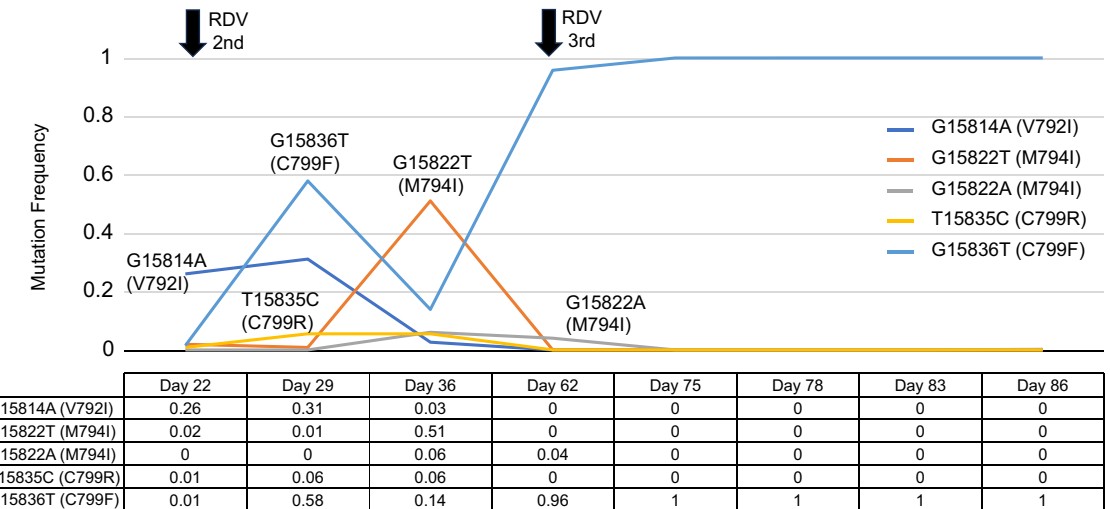

| | Day 22 | Day 29 | Day 36 | Day 62 | Day 75 | Day 78 | Day 83 | Day 86 |
|---|---|---|---|---|---|---|---|---|
| G15814A (V792I) | 0.26 | 0.31 | 0.03 | 0 | 0 | 0 | 0 | 0 |
| G15822T (M794I) | 0.02 | 0.01 | 0.51 | 0 | 0 | 0 | 0 | 0 |
| G15822A (M794I) | 0 | 0 | 0.06 | 0.04 | 0 | 0 | 0 | 0 |
| T15835C (C799R) | 0.01 | 0.06 | 0.06 | 0 | 0 | 0 | 0 | 0 |
| G15836T (C799F) | 0.01 | 0.58 | 0.14 | 0.96 | 1 | 1 | 1 | 1 |

**Fig. 2 | Emergence and selection of RDV resistant mutations in SARS-CoV-2.** On Day 22, an Omicron BA.5 variant, termed CH-LT1, and a V792I mutant in nsp12, termed CH-LT1m, were isolated; on Day 29, a C799F mutant in nsp12, termed CH-LT2m, and on Day 36, M794I (M794I(G15822T) and M794I(G15822A) mutants, termed CH-LT3m, were identified. The viral isolate obtained on Day 62 (CH-LT4) had considerably diversified mutations, including C799F. The second course and third course of RDV was initiated on Day 23 and Day 62, respectively. RDV remdesivir.

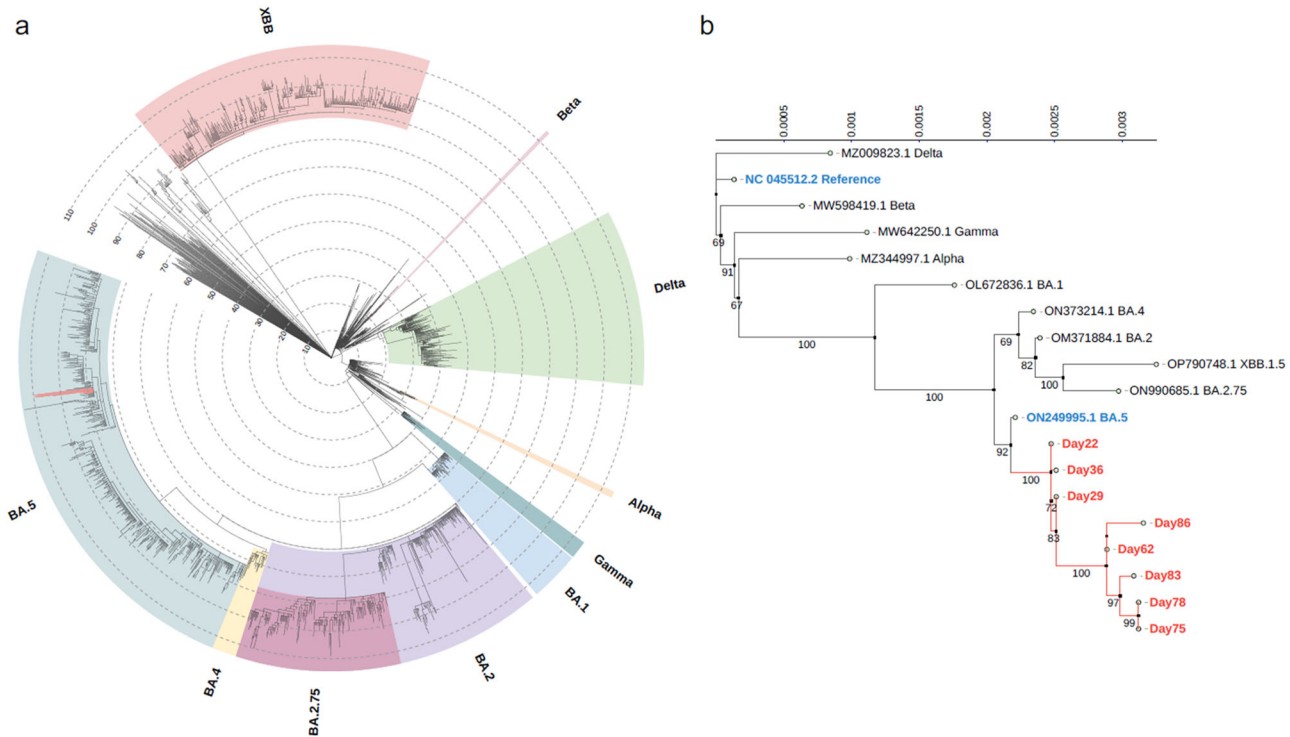

**Fig. 3 | Dynamic SARS-CoV-2 evolution occurred in a single patient during a short period. a** The lineage assignment by Nextclade. The scale number indicates number of mutations. The SARS-CoV-2 strains of the current case are highlighted with red nodes. **b** The inferred phylogenetic tree of current strains and VOC. The top scale and node numbers indicate distance and bootstrap value (%), respectively. The index strain of SARS-CoV-2 (NC_045512.2) and Omicron BA.5 (ON249995.1), and current strains are highlighted in blue and red, respectively.

have emerged from the second course. C799R was also identified on Days 29 and 36, both at a low frequency (5%). Perhaps because of this low frequency, we could not isolate the virus with the C799R mutation.

If limited to this patient, the emergence of the C799F mutation is of particular interest. Despite RDV treatment, this mutant virus could not be eliminated. We presume that the C799F mutation not only resulted in the acquisition of RDV resistance but also affected the fidelity of RdRp, thus causing the emergence of such dynamic

diversified mutations. In the case of HIV-1, relationships between mutational changes in reverse transcriptase and viral enzymatic fidelity have been reported[15,16]. Whether C799F alone affects viral RdRp fidelity or additional changes are required needs further investigation.

Viruses detected on Day 62, 1 day before the third course of RDV medication, exhibited evident fusogenicity. Similarly, the Delta variant is known to possess enhanced fusogenicity due to the P681R mutation, which is located near the furin cleavage site in the S protein.

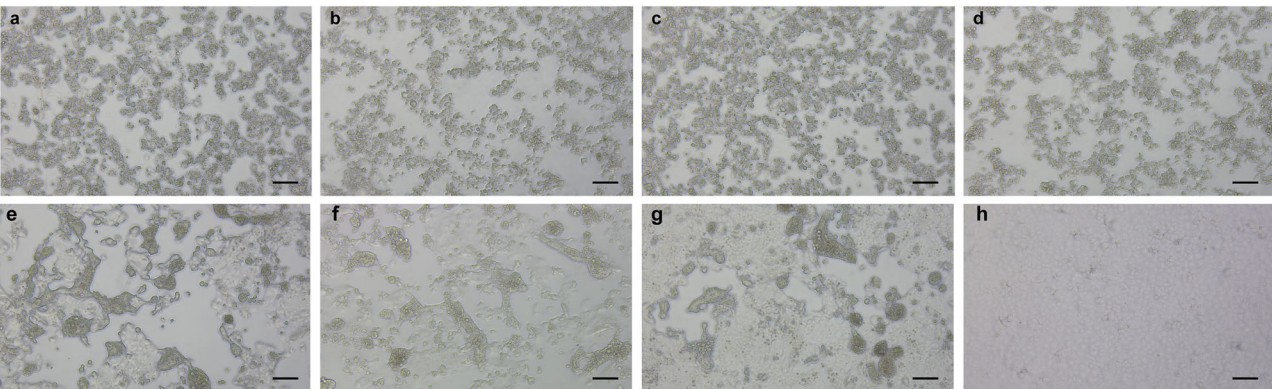

**Fig. 4 | SARS-CoV-2 BA.5 acquired fusogenicity during chronic persistent infection.** Cytopathogenic effects (CPEs) in VeroE6/TMPRSS2 caused by various isolates (CH-LT1, CH-LT1m, CH-LT2m, CH-LT3m, CH-LT4, CH-LT6, and CH-LT7) are shown. The isolates obtained on earlier days (from CH-LT1 to CH-LT3m) caused similar cell-rounding CPEs (**a-d**). Remarkably, the viral properties of CH-LT4 obtained on Day 62 appeared to change drastically and become more fusogenic (**e**). In addition to cell-rounding, large clumps of cells that merged with the adjacent cells were observed frequently. This fusogenic nature was maintained in the later isolates (CH-LT6 and CH-LT7, Days 78 and 83, respectively) (**f, g**). We performed three independent experiments, and obtained basically the similar results. Cell images were captured using a ZEISS Prim overt inverted microscope equipped with an Axiocam 208 colour camera at 3 d.p.i. Scale bars denote 100 μm. **a**: CH-LT1, isolated on Day 22, **b**: CH-LT1m, another variant with a V792I mutant in nsp12, isolated on Day 22, **c**: CH-LT2m, a variant with a C799F mutant in nsp12, isolated on Day 29, **d**: CH-LT3m, a variant with a M794I mutant in nsp12, isolated on Day 36, **e**: CH-LT4, a variant with C799F mutant in nsp12, isolated on Day 62, **f, g**: CH-LT6 and 7, variants with C799F mutant in nsp12, isolated on Day 78 and 83, respectively, **h**: control.

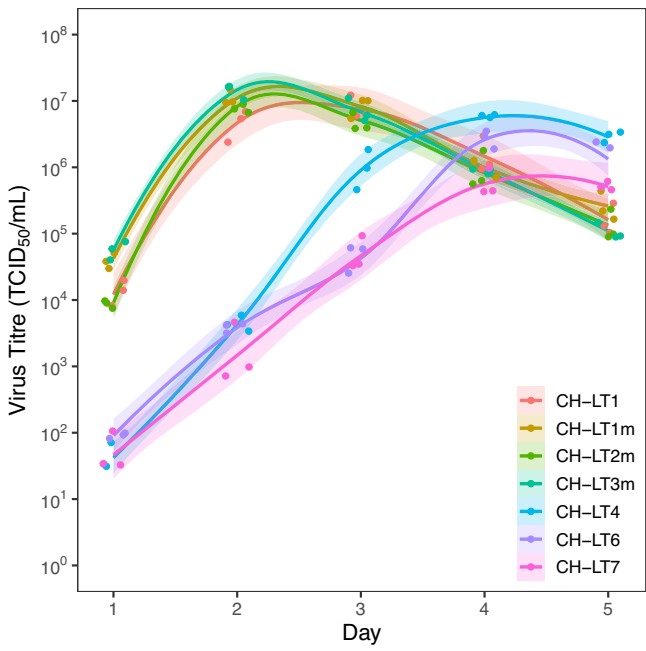

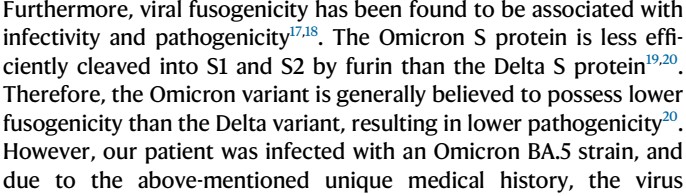

**Fig. 5 | The growth of SARS-CoV-2 was affected along with accumulating mutations.** Growth curves of the viral isolates obtained in the present study are shown. The growth of earlier isolates (from CH-LT1 to CH-LT3m) was relatively faster and reached the peak within 2 to 3 d.p.i., whereas that of later isolates (CH-LT4 to CH-LT7) was delayed 1–2 days and reached the peak at 4–5 d.p.i. The respective plot data of each isolate are defined by a key within the Figure. Lines represent mean values of three independent experiments, with 95% confidence intervals [mean ± 1.96 × SEM (standard error of mean)] as the shaded area.

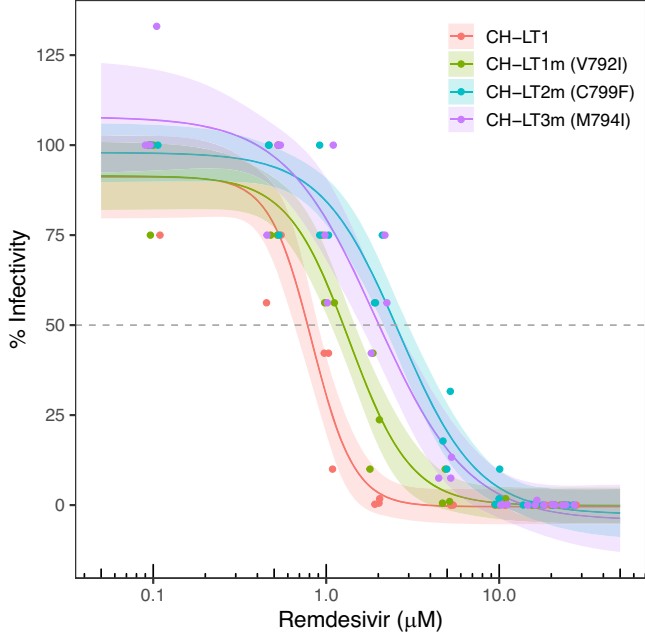

**Fig. 6 | Mutations in RdRp of SARS-CoV-2 resulted in RDV resistance.** Viral infectivity (%) was calculated from the viral titres (expressed as $TCID_{50}$/ml values) in the harvested culture supernatants 72 h after virus inoculation in the presence of various concentrations (ranging from 0.1 to 25 μM) of RDV divided by the titre without RDV (×100). Lines represent mean values of three independent experiments, with 95% confidence intervals [mean ± 1.96 × SEM (standard error of mean)] as the shaded area. RDV remdesivir.

Furthermore, viral fusogenicity has been found to be associated with infectivity and pathogenicity[17,18]. The Omicron S protein is less efficiently cleaved into S1 and S2 by furin than the Delta S protein[19,20]. Therefore, the Omicron variant is generally believed to possess lower fusogenicity than the Delta variant, resulting in lower pathogenicity[20]. However, our patient was infected with an Omicron BA.5 strain, and due to the above-mentioned unique medical history, the virus unexpectedly acquired fusogenicity during long-term persistent infection. The causes for the acquisition of fusogenicity remain unknown based on the currently available data. Position P681 remained as it was (P681H, the same as other BA.5 strains) throughout the whole clinical course (Table 3). Based on the increased Ct values and other clinical data, virus shedding appeared to stop after the third course of RDV treatment. The finally acquired viral fusogenicity possibly impacted the clinical course.

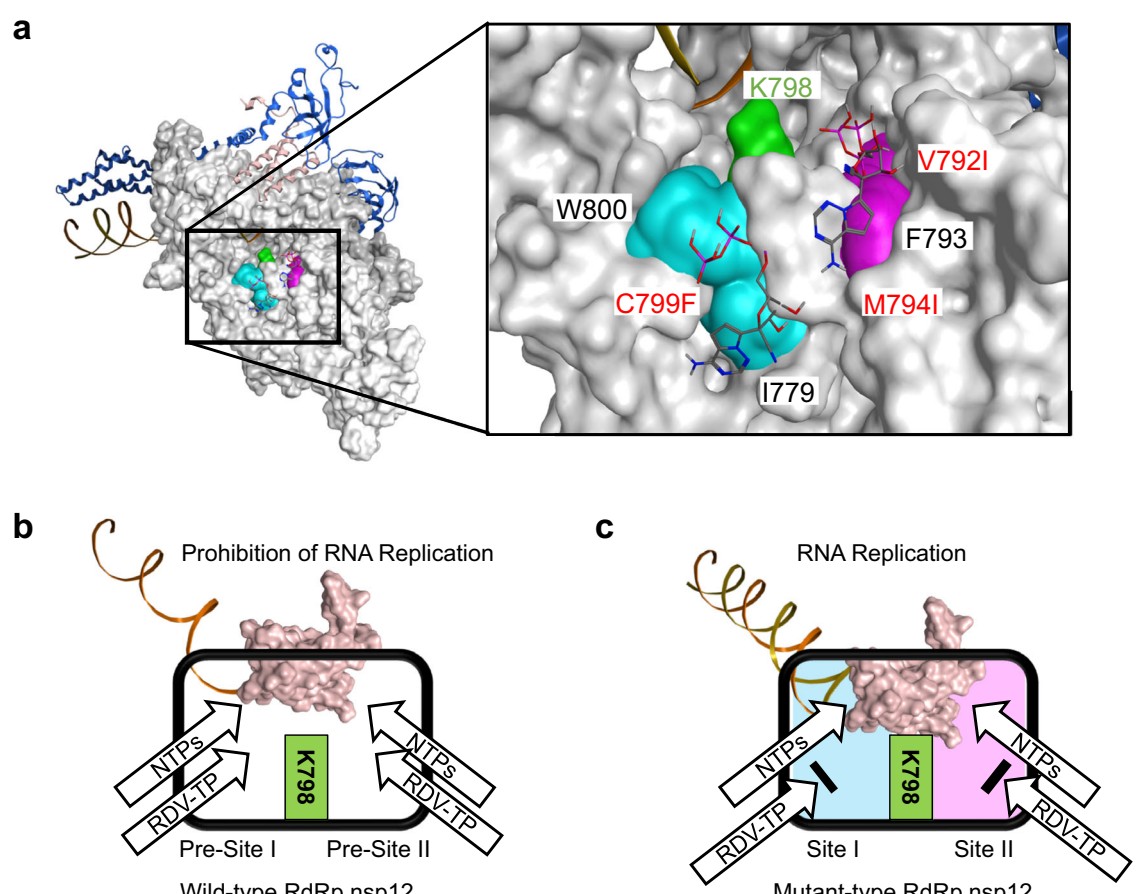

**Fig. 7 | in silico simulation of SARS-CoV-2 RdRp and schematical illustration.** **a** Predicted three-dimensional structure of mutant-type SARS-CoV-2 RdRp (V792I, M794I, and C799F) and expansion view of Domain-II region (after docking simulation of RDV-TP). Grey: RdRp, Cyan: Site-I (I779, C799F, and W800), Magenta: Site-II (V792I, F793, and M794I), Green: K798, Pink ribbon: nsp7, Blue ribbon: nsp8, Yellow ribbon: primer, Orange ribbon: template, Stick model: RDV-TP. Site I creates a large hydrophobic cluster by adding C799F to join the hydrophobic regions of I779 and W800, and Site II extends the hydrophobic cluster by V792I and M794I flanking the conserved hydrophobic residue F793. In addition, Sites I and II are located close to K798, presumed to play a pivotal role in the active site-linked polymerase activity of SARS-CoV-2 RdRp. **b** in silico molecular docking simulation suggested that RDV-PT incorporated into the active site of RdRp (Pre-site I: I779/ W800, Pre-site II: F793) of the index strain (NC_045512.2, Pango lineage B), and prevented the RNA replication. **c** RDV-TP was trapped through hydrophobic interactions with I779 (Site I: I779/C799F/W800) and F793 (Site II: V792I/F793/ M794I) and prevented from entering the polymerase active site of SARS-CoV-2 RdRp. RdRp RNA-dependent RNA polymerase, RDV-TP remdesivir-triphosphate.

In in vitro experiments, the EC$_{50}$ value of CH-LT2m, harbouring C799F and isolated on Day 29, was 2.7 μM (approximately three times higher than that (0.8 μM) of the parental virus, CH-LT1, which harbours no such mutation). Meanwhile, the EC$_{50}$ values of the other two mutants, CH-LT1m (V792I) and CH-LT3m (M794I), were 1.4 and 1.9 μM, respectively. We speculate that the acquisition of C799F may be a stochastic event. However, the highest RDV resistance was conferred by the C799F mutant, which may be the reason for the survival of C799F and the elimination of the other two mutants. As a result, the infectivity of the virus harbouring C799F was maintained to some extent, even in the presence of 5 μM or higher concentrations of RDV, allowing continued viral shedding with Ct values of 20–30 over a long time period.

To provide plausible explanations for our clinical and in vitro data, we investigated how SARS-CoV-2 RdRp acquired drug resistance to RDV-TP by using in silico simulation. We focused on the physicochemical characteristics of RDV-TP, an adenosine triphosphate (ATP) analogue with a 1′-C-nucleoside bond, 1′-ribose cyano, and a C7 substitution, which makes its hydrophobicity compared to ATP (Supplementary Fig. 3)[21]. In the case of the index strain of SARS-CoV-2 RdRp, we presumed that π-π stacking (weak hydrophobic interaction) attracts NTPs (ATP, UTP, GTP, CTP) and RDV-TP on I779 (Pre-site I) or F793 (Pre-site II), then inducing them to the active site (Fig. 7b)[14,22]. In

mutant SARS-CoV-2 RdRp, amino acid mutations at Site I (C799F) and Site II (V792I/M794I) enhanced the I779- and F793-mediated hydrophobic interaction with RDV-TP, respectively, preventing entry of RDV-TP into the active site (Fig. 7c)[7]. Thus, SARS-CoV-2 RdRp may have acquired drug resistance through increasing hydrophobic amino acids such as V792I, M794I, and C799F to flexible adaptation for the physicochemical property of RDV-RT hydrophobicity. Torii et al. showed that multiple mutations in nsp12 affected the RNA-binding site flexibility and decreased the antiviral activity of RDV[23]. Our in silico simulation confirmed their conclusion.

The patient received the initial two doses of the SARS-CoV-2 mRNA vaccine and an additional booster dose, all based on the index strain of SARS-CoV-2. The increase in neutralising antibody titres against the Omicron BA.5 subvariant was lower compared to the response against the index strain of SARS-CoV-2[24]. In our case, the BA.5 subvariant may have evaded neutralising antibodies induced by these vaccinations. Considering the emergence of such mutant strains, the World Health Organization (WHO) addressed the need for vaccine composition updates, including the Omicron strain, in June 2022[25]. Reflecting on our case, the IgG titres against the S protein, measured at 142–190 U/ml after Day 75, were considerably lower than those observed in healthcare workers[26]. As such, the neutralising antibody titre against the BA.5 subvariant was expected to be limited even after

BA.5 infection. Considering both the attenuated immune condition post lung transplantation and the immunological escape of the BA.5 subvariant, these factors may have contributed to the severe clinical course observed in our patient.

Here, we reported a patient infected with SARS-CoV-2 during immunosuppressive therapy following lung transplantation. This case report highlights that inadequate treatment of immunosuppressed hosts can generate highly diverse SARS-CoV-2 mutants that threaten human health through the production of drug-resistant variants. The emergence of variants with dynamic mutations is an obstacle in developing effective vaccines and antiviral drugs. In the future, it will be vital to consider effective therapeutic strategies that do not generate mutant viral strains. We believe that the development of new RdRp inhibitors or combination therapy with other antiviral drugs targeting different viral components, such as 3CL protease, will be among future treatment options.

## Methods

### Ethics statement and patient consent
This study was approved by the Research Ethics Committees of Graduate School of Medicine, Chiba University (M10505). The instructions for patients include the following: Research purpose, research methods, expected effects and risks, not being disadvantaged even if you do not consent, being able to withdraw consent at any time after consent, strict management of personal information, research results being reported in academic journals, research organization and funding sources. Participant gave written informed consent, according to CARE guidelines and in compliance with the Declaration of Helsinki principles.

### RDV treatment
The patient received three courses of RDV medication. During each course, the patient received an initial dose of 200 mg IV, followed by four daily doses of 100 mg IV (5 days in total).

### Detection of SARS-CoV-2 RNA
SARS-CoV-2 RNA was detected using a real-time RT-PCR kit (Ampdirect® 2019-nCoV detection kit; Shimadzu, Kyoto, Japan).

### Whole-genome sequencing
Reverse transcription was performed using a LunaScript® RT SuperMix Kit (New England Biolabs, MA, USA) as the following cycling condition: 25 °C, 2 min; 55 °C, 10 min; 95 °C, 1 min. Then, a 2 kb tiling PCR was performed using a standard protocol with Tks Gflex® DNA polymerase (Takara Bio, Shiga, Japan) and four primer pools (Supplementary Table 2, synthesised by Integrated DNA Technologies, IA, USA) as the following cycling condition: 94 °C, 1 min; 40 cycles at 98 °C, 10 s, 60 °C, 15 s, and 68 °C, 1 min. After amplification, a library was prepared using the xGen® DNA Library EZ UNI Kit (Integrated DNA Technologies, IA, USA) and sequenced using an iSeq100 instrument (Illumina, CA, USA). Sequencing data were pre-processed using fastp (trimming 1 base in 5′- and 3′-ends of reads)[27] and mapped onto the SARS-CoV-2 genome (NC_045512) using a BWA aligner[28]. Trimming of primer sequences, variant calling, and consensus sequence generation were performed using iVar with default settings (variants with ≥3% frequencies were called)[29]. Used commands are shown in Supplementary information. WGS (Whole Genome Sequencing) data are available in the NCBI Sequence Read Archive (SRA), submission SUB13521440 under BioProject number PRJNA983865 (BioSample accession numbers SAMN35736960–SAMN35736967, https://www.ncbi.nlm.nih.gov/bioproject/?term=PRJNA983865).

### Phylogenetic analysis
SARS-CoV-2 lineage was assigned by both Pangolin (https://cov-lineages.org/index.html)[30] and Nextclade (https://clades.nextstrain.org/)[31] web applications. In addition, the phylogenetic analysis was performed using whole genome sequence data of current strains and consensus sequence of WHO designated VOC, Alpha, Beta, Gamma, Delta, and Omicron (BA.1, BA.2, BA.4, BA.5, BA.2.75, and XBB.1.5) variants. The multiple sequence alignment was performed by MAFFT[32]. The phylogenetic tree was inferred by IQ-TREE with 1000 bootstrap resampling[33,34]. The best-fit substitution model (TIM + F + I) was selected by ModelFinder plus option[35]. The inferred phylogenetic tree was visualised by iTOL (https://itol.embl.de/) web application[36].

### Cells and virus isolation
TMPRSS2-expressing VeroE6 cells (JCRB1819), a SARS-CoV-2-susceptive cell line[11], were obtained from the JCRB (Japanese Collection of Research Bioresources Cell Bank, Tokyo, Japan), and cultured in Dulbecco's modified Eagle medium-high glucose (DMEM; Sigma-Aldrich, London, UK) supplemented with 5% (v/v) fetal bovine serum (FBS; Cytiva, Tokyo, Japan) and an antibiotic mixture containing penicillin G (100 units/ml), streptomycin (100 μg/ml), and amphotericin B (0.25 μg/ml) (Nacalai Tesque, Kyoto, Japan). Viral isolation was achieved by inoculating a portion of a saline extract of a nasopharyngeal swab from the patient (Day 22 and thereafter) onto a VeroE6/TMPRSS2 culture in a 6-well plate. The cultures were incubated at 37 °C/5% $CO_2$ and monitored by daily microscopic observation.

### Preparation of stock virus and cytopathogenic effect (CPE)
Conspicuous CPE (usually cell rounding) spread throughout the well of the culture plate, usually 2–4 d.p.i. After low-speed centrifugation ($800 \times g$, 6 min), the culture supernatant was harvested and stored at −70 °C as isolated virus stock. For further experiments, a working virus solution was prepared by inoculating a portion of the stock onto fresh VeroE6/TMPRSS2 cells cultured in a 25-cm² flask. The cells were cultured for 2–3 days, and the supernatants were harvested when full-blown CPE was observed and stored in the same manner as the original stock.

### Growth kinetics of the viral isolates
To examine the viral growth properties of respective isolates, we performed the following experiments in triplicate. First, VeroE6/TMPRSS2 cells were seeded in 6-well plates (4 ml/well) and allowed to nearly reach confluence within 2 days. The medium was removed, and 1 ml of the new medium was added. Then, 100 μl of virus solution adjusted to a multiplicity of infection (MOI) of 0.01 (~4000 $TCID_{50}$ of each virus / ($4 \times 10^5$ cells per well)) was inoculated, and the plates were placed in a $CO_2$ incubator for 1 h, with occasional gentle shaking. This virus inoculum solution was removed and washed once with 2 ml of the new medium, and 4 ml of the fresh medium was added again. These were cultured further at 37 °C/5% $CO_2$. Aliquots of the culture supernatants were harvested every 24 h for 5 days, and their viral titres were quantified as described below.

### RDV resistance assay
To investigate the extent of RDV resistance in the various isolates, inhibitor assessment experiments were performed according to the method described by Stevens et al.[12]. First, VeroE6/TMPRSS2 cells were seeded in 24-well plates (1 ml/well) and allowed to nearly reach confluence within 2 days. The medium was removed, and 500 μl of the new medium containing various concentrations of RDV (GS-5734; Aobious, MA, USA), ranging from 0.1–25 μM, was added. Then, 100 μl of the test specimen virus solution (CH-LT1 to CH-LT3m) was added into each well at a MOI of 0.001 (~100 $TCID_{50}$ of each virus / ($10^5$ cells per well)), and the plates were placed in a $CO_2$ incubator for 1 h with occasional gentle shaking. This virus inoculum solution was removed and washed once with 500 μl of the RDV-containing medium, and 1 ml of the medium containing various concentrations of RDV was added again. These were cultured further at 37 °C/5% $CO_2$. Culture

supernatants were harvested 72 h after virus inoculation, and viral titres were quantified. Viral infectivity (%) was calculated from the viral titres (expressed as $TCID_{50}$/ml values) in the harvested culture supernatants 72 h after virus inoculation in the presence of various concentrations (ranging from 0.1 to 25 μM) of RDV divided by the titre without RDV (×100). Dose-response analysis and calculation of median effect concentration ($EC_{50}$) values were performed using the "drc" package (version 3.0-1) from the R statistical software (version 4.2.2)[37].

We did not include CH-LT4 and later isolates in this assay because their growth curves were different from those of earlier isolates. We wanted to focus on the impact of mutation(s) that occurred in nsp12. Alternately, we separately performed the comparison experiments of later isolates with the original isolate CH-LT1 in the presence or absence of 5 μM RDV at 3 and 4 d.p.i. We used 6-well plates, and the MOI was adjusted to 0.01. Other conditions were the same as described above. All RDV resistance experiments were performed in triplicate and statistical analysis was treated.

#### Quantification of viral infectious titre
VeroE6/TMPRSS2 cells were seeded in 96-well plates (100 μl/well) in a similar manner as described in the growth kinetics experiment, and allowed to nearly reach confluence within 2 days, with 100 μl of 10-fold serial dilutions of virus-containing culture supernatants added into each well. The presence of live virus in each well was determined based on the CPE at 4 d.p.i., and the $TCID_{50}$ values were calculated using the Behrens–Kärber method. The viral titres are expressed as the $TCID_{50}$/ml.

#### Serological reactivity to SARS-CoV-2 S and N proteins
Antibody responses against the S and N proteins were analyzed using Anti-SARS CoV-2 S RUO and Elecsys® Anti-SARS-CoV-2 RUO (Roche Diagnostics, Switzerland), respectively, on the Cobas 8000 e801 module (Roche Diagnostics). The former system allows for the quantitative detection of antibodies, predominantly IgG, that target the viral S protein receptor-binding domain. The measurement threshold is 0.4 U/ml, and values of ≥0.8 U/ml were considered positive. The latter system allows for the quantitative detection of antibodies targeting the viral N antigen, with values of ≥1.0 considered positive.

#### in silico analysis
The three-dimensional structure of the SARS-CoV-2 RNA-dependent RNA polymerase (SARS-CoV-2 RdRp) harbouring V792I, M794I, and C799F mutations at Domain-II was constructed with MOE, version 2022.02 (CCG Inc, Montreal, Canada), based on the Brookhaven Protein Databank 6XEZ. Docking simulations of SARS-CoV-2 RdRp of V792I, M794I, and C799F mutants with RDV-TP (PubChem CID 56832906) were performed using the Amber99 force field in MOE.

#### Software for analysis
All the software(s) used in our study are freely available through the following sites except Molecular Operating Environment (MOE).

**Whole-genome sequencing.** fastp (version 0.23.2, https://github.com/OpenGene/fastp).
bwa (version: 0.7.17-r1188, https://github.com/lh3/bwa).
iVar (version 1.4, https://andersen-lab.github.io/ivar/html/index.html).

**Phylogenetic analysis.** MAFFT (version 7, https://mafft.cbrc.jp/alignment/software/).
IQ-TREE (version 2.2.6, http://www.iqtree.org).
iTOL (version 6.8.1, https://itol.embl.de).

**Dose-response analysis for RDV resistance experiment.** R (version 4.2.2, https://www.r-project.org).

drc package (version 3.0-1, https://cran.r-project.org/web/packages/drc/index.html).

**Three-dimensional structure analysis.** Molecular Operating Environment (MOE), version 2016.08 (CCG Inc, Montreal, Canada) is commercially available (https://www.chemcomp.com/index.htm).

#### Reporting summary
Further information on research design is available in the Nature Portfolio Reporting Summary linked to this article.

## Data availability
WGS data of SARS-CoV-2 in nasopharyngeal samples (Day 22, Day 29, Day 36, Day 62, Day 75, Day 78, Day 83, and Day 86) are uploaded on the following publicly available repository. Consensus sequence data are available from GISAID: the accession numbers, EPI_ISL_18261614–EPI_ISL_18261621. In addition, fastq files were also uploaded to NCBI Sequence Read Archive: BioProject number, PRJNA983865 (BioSample accession numbers, SAMN35736960–SAMN35736967). Source data are provided with this paper.

## Code availability
In this study, all the analyses were performed using the softwares described in the "Method" section. Analyses codes were provided in the Supplementary Information. There is no newly developed software.

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

## Acknowledgements

This study was funded by the Research Program on Emerging and Re-emerging Infectious Diseases (Project title: Chiba University, Synergy Institute for Futuristic Mucosal Vaccine Research and Development) funded by Japan Agency for Medical Research and Development (AMED) (JP223fa627003).

## Author contributions

H.I. is the primary author of the article. H.I., T.I., K.S., and E.I. drafted the manuscript and performed the literature search. S.S., M.Y., T.T., and T.S. were a part of the infectious diseases team that provided direct clinical care to the patient. T.I., S.M., and K.M. conducted genome-wide sequencing analysis, qRT-PCR, and serological tests. E.I. and K.S. conducted virus isolation and RDV resistance assay. Y.T. and A.S. performed in silico analysis. All authors participated in manuscript revision, agreed to submit the manuscript, and approved the final version of the manuscript. All clinical authors had full access to the clinical data.

## Competing interests

The authors declare no competing interests.
