## [Peer Review File · Nature Communications]

REVIEWER COMMENTS

Reviewer #1 (Remarks to the Author):

In this study, Igari et al. present a clinical case study investigating an immunosuppressed patient - a recipient of lung transplantation - who, despite being treated with RDV, developed a chronic, persistent SARS-CoV-2 infection. The authors further probe into various genetic mutations in the virus, with a particular focus on the C799F mutation in the nsp12 viral RNA polymerase, which appears to promote resistance to RDV. Building upon previous studies on chronic infections in immunocompromised individuals, this work underscores the complexities of managing COVID-19 in these patient populations, the potential of viral mutations to foster drug resistance, and the ongoing need for vigilant surveillance and evolving treatment strategies.

To strengthen this succinct, well-written study, I would suggest some following changes be implemented:

Results:

1. Lines 62-63: Despite the case of a vaccinated individual contracting COVID-19, it's essential to note that early vaccines targeted the Spike protein of the initial wild-type strain, meaning this observation is not unusual.
2. Lines 80-91: The mutations plot in Figure 1 does not contribute significantly to understanding the specific Remdesivir-associated mutations the authors are describing in the manuscript. Instead of showing total mutations, additionally tracking the V729I, C799F, M749I, and C799F trajectories would be more insightful rather than reading from a quite long table.
3. The addition of a phylogenetic tree would illustrate virus evolution at each time point more explicitly, potentially revealing significant viral population shifts.
4. In table 1 the authors show clinical course of the patient – interesting is the high amount of inflammation seen at days 36-62. Perhaps a figure (or add these to make figure 1a/1b?) of these parameters might be better than a table

5. Figure 4 is not hugely intuitive in what it is showing for me. Perhaps some annotation of interaction of the ligand interactions may prove more informative.

Discussion/Methods:

1. Line 151: The detection of a mutation at consensus frequency at a singular time point, followed by its disappearance, warrants further discussion (M749I). This pattern suggests strong selection pressure, poor fitness, or compartmentalization.

2. Lines 159/220: The use of iVar with default settings for mutation calling may produce varying results. More explicit methodological information is needed, including the specific read depth and criteria for variant calling.

3. Line 165: Instead of speculation, pseudotyped virus experiments could provide a direct assessment of the impact of single versus dual mutations on Remdesivir in vitro.

4. The number of replicates for each experiment should be specified for replication and robust statistical analysis. I would expect at least three replicates for fusogenicity and resistance assays, but I cannot see this in the text.

5. More context regarding the choice for the RDV treatment regimen would provide a clearer understanding of the study. For example, why was paxlovid or other drugs not attempted?

6. For the whole-genome sequencing, the section could provide details about how the sequencing data was analysed beyond just mapping, trimming, and variant calling. The authors state that iVar was used with default settings, but it may be more useful to know more details about how the sequence reads were pre-processed, aligned, what specific criteria were used for variant calling. I would strongly suggest additional downstream analyses be performed including phylogenetic analysis and comparison to other sequenced SARS-CoV-2 genomes

7. You may find that dropping your threshold further to e.g. 2% may allow you to track low-minority variants, rather than concluding that it was missing entirely. Indeed, even haplotype reconstruction could be used to identify viral populations with this mutation.

8. Further clarification is needed for the monitoring of cells during virus isolation and the subsequent quantification of changes.

9. For the Remdesivir resistance assay, how viral titers were quantified needs to be specified.

10. The serological reactivity analysis could benefit from additional information about how results were interpreted, and any steps taken to ensure specificity.

11. The in silico analysis would benefit from more details on the criteria for docking success and how results were interpreted.

General suggestions:

1. Depositing all consensus sequences and/or fastq files in an open-source repository would enable further analysis by other researchers and to ensure that mutations and tracked variants are reproducible – indeed I cannot find the C799F reported in many other sequences in the GISAID repository, so this may be a crucial finding!

2. The study only cites two other clinical cases of immunosuppressed patients. Another relevant case was published by Kemp et al., which should be acknowledged (<https://www.nature.com/articles/s41586-021-03291-y>).

Reviewer #2 (Remarks to the Author):

Igari et al describe a case of COVID-19 in a lung transplant recipient that persisted despite multiple courses of remdesivir. The authors performed whole-genome sequencing of the virus at various timepoints, and showed that a large number of mutations developed, including several nsp12 mutations that conferred some level of resistance in vitro to remdesivir. Specifically, they show that

the C799F mutation resulted in increased fusogenicity, a property that has been shown in other strains to result in increased viral pathogenicity. They conclude that inadequate antiviral treatment in immunocompromised patients may result in development of new viral strains that threaten public health.

The paper is well-written and generally well-organized, and the problem of prolonged infection and failure of antiviral treatments in immunocompromised patients is an important one. From a clinical standpoint, the report is not clearly novel. There have been numerous case reports of similar clinical courses, as well as characterizations of nsp12 mutations that cause RDV resistance (eg, Hogan et al, as cited by the authors). Prior papers have also demonstrated the accumulation of many diverse mutations in immunocompromised patients over time (for example – PMIDs 33176080 and 35996593). It has also been widely postulated that immunocompromised patients represent a potential concerning site of viral evolution for COVID-19, so this conclusion is not novel. The paper does show quite elegantly that the C799F mutation developed in vivo and appeared to cause increased fusogenicity to arise in an Omicron strain; to my knowledge, this has not been previously demonstrated. It would be helpful for the authors to state more explicitly in both the abstract and the conclusion exactly what is novel about this work.

From the standpoint of the clinical case, I would suggest that the authors include the patient's exact immunosuppressive regimen, as well as a mention of whether or not the patient had recently received augmented immunosuppression for rejection or rheumatologic flare. It would be helpful to include explicitly during the case description the fact that cycle threshold values remained low throughout the patient's course, providing evidence that the patient's ongoing illness was in fact due to ongoing SARS-CoV-2 replication.

Reviewer #3 (Remarks to the Author):

The authors reported a clinical case of emergence of remdesivir (RDV)-resistant Omicron BA5 variants in an immunosuppressed patient who received a lung transplant. The authors monitored the number and frequency of mutations in the whole viral genome of nasal specimens collected through the clinical course. Four isolates termed CH-LT1m (isolated on day 22), CH-LT2m (on day 29), CH-LT3m (on day 36), and CH-LT4m (on day 62), possessing mutations in nsp12 (i.e., V792I, C799F, and M794I), as well as in other ORFs, were presented and characterized. Of these mutations in nsp12, only C799F continued being detected after the viruses were isolated.

This is a key clinical case report; however, the authors' findings and suggestion do not seem advantageous and there are several concerns that need be addressed in the current manuscript.

1. The authors stated that “The amino acid mutations compared to those in the confirmed BA.5.2.1 genome were categorized as major or minor...,” (lines 82–84), but the frequency and the number of mutations described in the result section, Figure 1, Table 2, and Table 3 were compared to those in the Wuhan strain. This may be misleading or confusing to readers. Accurate numbers compared to the BA5 variant should be clearly provided. Indeed, the number of major mutations (>50% frequency) that were expected due to RDV treatment was around seven (Table 3).
2. Amino acid sequence information on the isolates (CH-LT1m, CH-LT2m, CH-LT3m, and CH-LT4m) was not provided. Mutations relevant to the BA5 variant and compared to CH-LT1 should be listed in a table.
3. Regarding the CPE in Figure 2, MOI, Days post-infection, images of other isolates, and the sequence information of the spike gene should be provided. Two isolates (Days 78 and 83) with high fusogenic activity were introduced without showing data or details in lines 109–110. These isolates seem interesting as well.
4. The authors stated in the abstract that “the mutation C799F in nsp12 was found to confer RDV resistance”, but direct evidence was not shown in the manuscript although the contribution of the C799F mutation to RDV resistance was suggested in the paper to which the authors referred (Ref #17).
5. The authors performed the RDV resistance assay with three isolates (CH-LT1m, CH-LT2m, and CH-LT3m), showing slightly high EC50 values (1.75-fold, 3.38-fold and 2.38-fold increases, respectively). Although the other isolate CH-LT4m has the same C799F mutation as CH-LT2m that could confer RDV resistance, CH-LT4m was not included in this assay. Why not? It should be included.
6. Statistical analysis was not provided for the RDV resistance assay.
7. To validate the RDV resistance of the isolates, growth kinetics in the presence and absence of RDV treatment should be provided.
8. IgG antibody titers against SARS-CoV-2 S and N, described in lines 63–65, should be provided along with the Ct value in Table 1.
9. In Table 1, the Ct value on Day 78 is “2.4”. This must be incorrect.
10. In lines 86 and 89, “20 (11+9) locations” and “50 (21+29) locations”. An explanation of what the numbers in parentheses indicate is needed.

Point by Point Response to Reviewer Comments

Manuscript ID; NCOMMS-23-23503

Reviewer #1 (Remarks to the Author):

In this study, Igari et al. present a clinical case study investigating an immunosuppressed patient - a recipient of lung transplantation - who, despite being treated with RDV, developed a chronic, persistent SARS-CoV-2 infection. The authors further probe into various genetic mutations in the virus, with a particular focus on the C799F mutation in the nsp12 viral RNA polymerase, which appears to promote resistance to RDV. Building upon previous studies on chronic infections in immunocompromised individuals, this work underscores the complexities of managing COVID-19 in these patient populations, the potential of viral mutations to foster drug resistance, and the ongoing need for vigilant surveillance and evolving treatment strategies.

To strengthen this succinct, well-written study, I would suggest some following changes be implemented:

Response to Reviewer #1

We thank the reviewer for these very positive comments on the work. We provide a point-to-point response to every question raised.

Results:

Comment 1

Lines 62-63: Despite the case of a vaccinated individual contracting COVID-19, it's essential to note that early vaccines targeted the Spike protein of the initial wild-type strain, meaning this observation is not unusual.

Response to Comment 1

We thank the reviewer for this comment. We completely agree that the immune status acquired via vaccination with early COVID-19 vaccines is reduced in an immunocompromised host, as in our case. As evidence to this, we measured IgG antibodies against the N and S proteins.

Therefore, the evidence presented in this sentence is usual and expected in an immunocompromised host. We have highlighted the immune status in the following sentence, with minor changes. "The patient's immunoglobulin G (IgG) titre against the SARS-CoV-2 spike (S) protein was elevated, while being very low compared to that after mRNA vaccination and SARS-CoV-2 infection. IgG against the nucleocapsid (N) protein was negative". These changes are highlighted in

the revised manuscript (Section of 'Clinical course and laboratory data' in Results).

Antibody titres were measured late in the clinical course, not immediately after admission. Please consider this in conjunction with Table 1.

Comment 2

Lines 80-91: The mutations plot in Figure 1 does not contribute significantly to understanding the specific Remdesivir-associated mutations the authors are describing in the manuscript. Instead of showing total mutations, additionally tracking the V729I, C799F, M749I, and C799F trajectories would be more insightful rather than reading from a quite long table.

Response to Comment 2

We thank the reviewer for this comment. We completely agree that amino acid mutations associated with RDV resistance are of great clinical interest. We prepared a new figure as **Figure 2** in the revised manuscript. We selected five nucleotide mutations (V792I (G15814A), M794I (G15822T), M794I (G15822A), C799R (T15835C), and C799F (G15836T)) associated with RDV resistance. The frequencies of these were graphed over time. Although NGS data indicated the presence of minor mutations, the viruses harbouring M794I (G15822A) or C799R (T15835C) could not be isolated perhaps due to their very low prevalence rates.

We added new descriptions, as highlighted in the revised manuscript (Section of 'Acquisition of RDV resistance via nsp12 mutations' in Results): “Meanwhile, CH-LT1m, CH-LT2m, and CH-LT3m, harbouring the V792I (G15814A), C799F (G15836T), and M794I (G15822T) mutations in nsp12, respectively, acquired some resistance against RDV”.

Comment 3

The addition of a phylogenetic tree would illustrate virus evolution at each time point more explicitly, potentially revealing significant viral population shifts.

Response to Comment 3

We thank the reviewer for this suggestion. We prepared a phylogenetic tree as the new **Figure 3**. The isolated viruses in this clinical course were confirmed to belong to BA.5.2.1 and revealed diverse dynamic changes within two months. This description is highlighted and found in the revised

manuscript (Section of 'Isolation of various SARS-CoV-2 mutants' in Results).

Comment 4

In table 1 the authors show clinical course of the patient – interesting is the high amount of inflammation seen at days 36-62. Perhaps a figure (or add these to make figure 1a/1b?) of these parameters might be better than a table.

Response to Comment 4

We thank the reviewer for this suggestion. We reconsidered the contents of both Table 1 and Figure 1, thus creating a new **Figure 1** with information on the clinical course. These changes are highlighted and found in the revised manuscript (Section of 'Clinical course and laboratory data' in Results).

Comment 5

Figure 4 is not hugely intuitive in what it is showing for me. Perhaps some annotation of interaction of the ligand interactions may prove more informative.

Response to Comment 5

We thank the reviewer for this comment. The previous Figure 4 did not sufficiently describe the phenomenon that causes RDV resistance. We prepared a new **Figure 7** and illustrated the mechanism of RDV resistance caused by C799F. These changes are highlighted and found in the revised manuscript (Section of '*in silico* simulation of SARS-CoV-2 RdRp' in Results).

Discussion/Methods:

Comment 1

Line 151: The detection of a mutation at consensus frequency at a singular time point, followed by its disappearance, warrants further discussion (M749I). This pattern suggests strong selection pressure, poor fitness, or compartmentalization.

Response to Comment 1

We thank the reviewer for this comment. Administration of RDV is a selective pressure for amino acid mutation. Information on amino acid mutations is provided in **Table 2, Table 3, Supplementary Table S1, and Supplementary Figure S1.**

Another reason for the phenomenon of quick disappearance of V792I and M794I could be ascribed to their viral instabilities, but we cannot explain from the present result.

Comment 2

Lines 159/220: The use of iVar with default settings for mutation calling may produce varying results. More explicit methodological information is needed, including the specific read depth and criteria for variant calling.

Response to Comment 2

The average read depth of each sample is shown in the Results section. Variant calling was also described in the Methods section.

Comment 3

Line 165: Instead of speculation, pseudotyped virus experiments could provide a direct assessment of the impact of single versus dual mutations on Remdesivir in vitro.

Response to Comment 3

Your suggestion to use pseudotyped virus (for example, VSV-based pseudovirus) would surely provide a direct assessment of the impact of single or several mutations if it were only in the Spike protein (outer membrane protein). In our case, we observed a few mutations in nsp12 (RdRp, one of the inner core proteins) which appeared to be crucial for RDV resistance. We judged that the use of isolated virus at each time point would be speedier and more suitable for obtaining the expected outcome instead of creating a new pseudotype virus system. We agree that the pseudotype virus experiments (where inner cores and outer membranes can be manipulated separately) would be very nice. However, performing them at this stage would be a challenge. We will consider these in future experiments.

Comment 4

The number of replicates for each experiment should be specified for replication and robust statistical analysis. I would expect at least three replicates for fusogenicity and resistance assays, but I cannot see this in the text.

Response to Comment 4

We thank the reviewer for this comment. Firstly, concerning the fusogenicity, we had observed the acquisition the fusogenicity in CH-LT4 and later isolates. To respond to the comment of reviewer 3 (comment 7), we performed the growth kinetics experiments in triplet. Taking advantage of this opportunity, we once again observed the changes of viral biological properties and captured images of all CPEs in all isolates. Thus, we modified **Figure 4** in the revised manuscript, in which the representative images of all isolates (CH-LT1 to CH-LT7) are shown together with control cells. Additional photos are available upon request.

Secondly, concerning the resistance assay, we performed it in triplicate as described in the Methods. However, only the mean values were plotted in the original figure to avoid too much complexity. To make this more explicit, we modified **Figure 6** in which the statistical analysis was performed in addition to plotting all the values. The legend of the figure was modified accordingly in the revised manuscript, as highlighted.

Comment 5

More context regarding the choice for the RDV treatment regimen would provide a clearer understanding of the study. For example, why was paxlovid or other drugs not attempted?

Response to Comment 5

We thank the reviewer for this comment. Paxlovid is known to interact with many drugs. Voriconazole, which is prescribed for the treatment of invasive pulmonary aspergillosis, is now listed as a contraindicated drug. For this reason, we did not use Paxlovid. Ensitrelvir was not available under the domestic regulatory review process, either.

Comment 6

For the whole-genome sequencing, the section could provide details about how the sequencing data was analysed beyond just mapping, trimming, and variant calling. The authors state that iVar was used with default settings, but it may be more useful to know more details about how the sequence reads were pre-processed, aligned, what specific criteria were used for variant calling. I would strongly suggest additional downstream analyses be performed including phylogenetic analysis and comparison to other sequenced SARS-CoV-2 genomes.

Response to Comment 6

We thank the reviewer for this comment. The used commands are shown in Supplementary Information. In addition, the phylogenetic analysis was also performed, as mentioned in 'Response to

Comment 3'.

Comment 7

You may find that dropping your threshold further to e.g. 2% may allow you to track low-minority variants, rather than concluding that it was missing entirely. Indeed, even haplotype reconstruction could be used to identify viral populations with this mutation.

Response to Comment 7

We agree with this comment. Minor variants with <3% frequency (V792I, M794I, and C799F) in nsp12 were tracked by counting read depth. The transitions of these mutations are shown in the new **Figure 2**.

Comment 8

Further clarification is needed for the monitoring of cells during virus isolation and the subsequent quantification of changes.

Response to Comment 8

We thank the reviewer for this comment. To describe biological changes in the isolated viruses more clearly, we modified **Figure 4** (showing the images of CPEs for all the isolates) in the revised manuscript. The switch to the fusogenic virus which occurred in CH-LT4 and later isolates (CH-LT6 and CH-LT7) can be easily observed in these images.

Comment 9

For the Remdesivir resistance assay, how viral titers were quantified needs to be specified.

Response to Comment 9

We thank the reviewer for this comment. The viral titres in the harvested culture supernatants were quantified as described in the sections of 'RDV resistance assay' and 'Quantification of viral titres' in Methods. The titres are expressed as TCID₅₀/ml. The percentages of inhibition were calculated based on these values. But this part how to calculate the percentage of inhibition had been described in the figure legend in the original manuscript as follows: Viral infectivity (%) was calculated from the viral titers in the harvested culture supernatants 72 hr after virus inoculation in the

presence of various concentrations (ranging from 0.1 to 25 μM) of RDV divided by the titer without RDV ($\times 100$).

To make the procedure more understandable, we modified the section of 'RDV resistance assay' in Methods by transferring the above-mentioned sentence in the resubmitted manuscript.

Comment 10

The serological reactivity analysis could benefit from additional information about how results were interpreted, and any steps taken to ensure specificity.

Response to Comment 10

We thank the reviewer for this comment. We completely agree. As was shown in **Table 1**, we examined the anti-spike protein IgG and anti-nucleocapsid IgG. The serum specimens from early days of the clinical course were not stored, and only the specimens collected in later days were measured. Despite vaccination and actual infection, the response was not sufficient. These changes are highlighted and found in the revised manuscript (Section of 'Clinical course and laboratory data' in Results).

Comment 11

The *in silico* analysis would benefit from more details on the criteria for docking success and how results were interpreted.

Response to Comment 11

We thank the reviewer for this comment. The criteria for docking success are based on binding state results (such as binding energies and binding conformation of RDV-TP on SARS-CoV-2 RdRp) via simulations and machine learning software.

The docking results are shown in the expanded view in **Figure 7** and are described in detail in the revised manuscript (Section of '*in silico* simulation of SARS-CoV-2 RdRp' in Results and the 8th paragraph in Discussion).

General suggestions:

Comment 1

Depositing all consensus sequences and/or fastq files in an open-source repository would enable further analysis by other researchers and to ensure that mutations and tracked variants are reproducible

– indeed I cannot find the C799F reported in many other sequences in the GISAID repository, so this may be a crucial finding!

Response to Comment 1

We thank the reviewer for this comment. All the consensus sequences of the virus isolates described in the present study in FASTA files were uploaded to GISAID, under accession numbers EPI_ISL_18261614–EPI_ISL_18261621. In addition, fastq files were uploaded to the NCBI Sequence Read Archive: BioProject number, PRJNA983865 (BioSample accession numbers, SAMN35736960–SAMN35736967).

These descriptions are highlighted and found in the revised manuscript (Section of ‘Data availability’ in Methods).

Comment 2.

The study only cites two other clinical cases of immunosuppressed patients. Another relevant case was published by Kemp et al., which should be acknowledged (<https://www.nature.com/articles/s41586-021-03291-y>).

Response to Comment 2

We thank the reviewer for this comment. We completely agree. A paper published in Nature by Kemp SA et al. has been added and found in the section of ‘Introduction’ in the revised manuscript (Reference number 9).

Reviewer #2 (Remarks to the Author):

Igari et al describe a case of COVID-19 in a lung transplant recipient that persisted despite multiple courses of remdesivir. The authors performed whole-genome sequencing of the virus at various timepoints, and showed that a large number of mutations developed, including several nsp12 mutations that conferred some level of resistance in vitro to remdesivir. Specifically, they show that the C799F mutation resulted in increased fusogenicity, a property that has been shown in other strains to result in increased viral pathogenicity. They conclude that inadequate antiviral treatment in immunocompromised patients may result in development of new viral strains that threaten public health.

The paper is well-written and generally well-organized, and the problem of prolonged infection and failure of antiviral treatments in immunocompromised patients is an important one. From a clinical standpoint, the report is not clearly novel. There have been numerous case reports of similar clinical courses, as well as characterizations of nsp12 mutations that cause RDV resistance (eg, Hogan et al, as cited by the authors). Prior papers have also demonstrated the accumulation of many diverse mutations in immunocompromised patients over time (for example – PMIDs 33176080 and 35996593). It has also been widely postulated that immunocompromised patients represent a potential concerning site of viral evolution for COVID-19, so this conclusion is not novel. The paper does show quite elegantly that the C799F mutation developed in vivo and appeared to cause increased fusogenicity to arise in an Omicron strain; to my knowledge, this has not been previously demonstrated. It would be helpful for the authors to state more explicitly in both the abstract and the conclusion exactly what is novel about this work.

From the standpoint of the clinical case, I would suggest that the authors include the patient's exact immunosuppressive regimen, as well as a mention of whether or not the patient had recently received augmented immunosuppression for rejection or rheumatologic flare. It would be helpful to include explicitly during the case description the fact that cycle threshold values remained low throughout the patient's course, providing evidence that the patient's ongoing illness was in fact due to ongoing SARS-CoV-2 replication.

Response to Reviewer #2

We thank the reviewer for these comments. We completely agree. We prepared a new modified **Figure 1**. The new **Figure 1** contains the changes in WBC and CRP, as inflammatory responses, and drugs for COVID-19 and associated conditions. The description of the clinical course summarised in **Figure 1** is highlighted and found in the revised manuscript (Section of 'Clinical course and laboratory data' in Results).

As pointed out by the reviewer, there are reports that immunosuppressed hosts develop chronic persistent infection, and that RDV-resistant viruses emerge from mutations in nsp12. However,

fusogenicity, which was confirmed in the Delta strain, has not been observed in the Omicron strain. In our case, fusogenicity, which was not seen at first, was observed in the clinical course. It appears that SARS-CoV-2 has acquired fusogenicity through accumulated mutations. We consider that our revised manuscript shows this very interesting phenomenon in a clearer manner than the original version. Although we have not determined the mechanism underlying the acquisition of viral fusogenicity in the Omicron strain, we believe that the results provided herein would be a good basis for further research.

Reviewer #3 (Remarks to the Author):

The authors reported a clinical case of emergence of remdesivir (RDV)-resistant Omicron BA5 variants in an immunosuppressed patient who received a lung transplant. The authors monitored the number and frequency of mutations in the whole viral genome of nasal specimens collected through the clinical course. Four isolates termed CH-LT1m (isolated on day 22), CH-LT2m (on day 29), CH-LT3m (on day 36), and CH-LT4m (on day 62), possessing mutations in nsp12 (i.e., V792I, C799F, and M794I), as well as in other ORFs, were presented and characterized. Of these mutations in nsp12, only C799F continued being detected after the viruses were isolated.

This is a key clinical case report; however, the authors' findings and suggestion do not seem advantageous and there are several concerns that need be addressed in the current manuscript.

Response to Reviewer #3

We thank the reviewer for these comments. We completely agree. We have revised the manuscript thoroughly as much as possible to respond to all the reviewer's comments.

Comment 1

The authors stated that "The amino acid mutations compared to those in the confirmed BA.5.2.1 genome were categorized as major or minor...", (lines 82–84), but the frequency and the number of mutations described in the result section, Figure 1, Table 2, and Table 3 were compared to those in the Wuhan strain. This may be misleading or confusing to readers. Accurate numbers compared to the BA5 variant should be clearly provided. Indeed, the number of major mutations (>50% frequency) that were expected due to RDV treatment was around seven (Table 3).

Response to Comment 1

We thank the reviewer for this comment. We moved the previous Table 3 to Supplementary and listed as **Table S1**. We prepared new **Table 3** which concisely show the mutations in nsp12 and spike protein. Then, we prepared the new **Figure 3** which is a phylogenetic tree of the BA.5.2.1 group of SARS-CoV-2, including the isolates obtained in this clinical course. By this phylogenetic tree, the relationships among all the isolates can be easily seen.

Comment 2

Amino acid sequence information on the isolates (CH-LT1m, CH-LT2m, CH-LT3m, and CH-LT4m) was not provided. Mutations relevant to the BA5 variant and compared to CH-LT1 should be listed in a table.

Response to Comment 2

We thank the reviewer for this comment. We prepared phylogenetic tree as the new **Figure 3**. The isolated viruses in this clinical course were confirmed to belong to BA.5.2.1 and revealed diverse dynamic changes over two months. This description is highlighted and found in the revised manuscript (Section of 'Isolation of various SARS-CoV-2 mutants' in Results).

As you pointed out, it is important to mention the final sequence of the respective isolates apart from the genome-wide sequence data obtained via Illumina. To confirm whether each isolate possesses the mutations indicated via NGS data, we performed RT-PCR analysis of nsp12 and S-gene regions (only these two are unstably drifting regions) and found that the isolates harboured the exact consensus sequences indicated by the NGS analysis, including mutations such as C799F, etc. The consensus sequences are deposited in the 'GISAID.' We have noted this in the revised manuscript (Section of 'Data availability' in Methods).

Comment 3

Regarding the CPE in Figure 2, MOI, Days post-infection, images of other isolates, and the sequence information of the spike gene should be provided. Two isolates (Days 78 and 83) with high fusogenic activity were introduced without showing data or details in lines 109–110. These isolates seem interesting as well.

Response to Comment 3

We thank the reviewer for this comment. CPE images in the original Figure 2 were taken at 3 d.p.i., as described in the Result section. We agree that we should show the CPE images of other isolates. Although we possess images of other isolates, we decided to perform the growth kinetics experiments of the respective isolates as you requested. Taking advantage of this opportunity, we once again attempted to observe the changes in viral properties and capture CPE images. We have included them in the new **Figure 4** in the revised manuscript showing one of the representative images for each isolate. The MOI in this additional experiment was 0.01 (approximately 10 times higher than that (0.001) of the RDV resistance assays) because the growth(s) of CH-LT4 and later isolates would be much delayed if we adopt the MOI of 0.001. All the pictures were taken at 3 d.p.i. We described these facts in the revised manuscript (**Section of 'Switch to fusogenic strains' in Results**).

Concerning the consensus sequence data, including the Spike regions, of the respective isolated viruses, we uploaded them on 'GISAID' We have noted this in the revised manuscript (**Section of 'Data availability' in Methods**).

Comment 4

The authors stated in the abstract that “the mutation C799F in nsp12 was found to confer RDV resistance”, but direct evidence was not shown in the manuscript although the contribution of the C799F mutation to RDV resistance was suggested in the paper to which the authors referred (Ref #17).

Response to Comment 4

Thank you for this comment. We think that the data of the new **Figure 6** (original Figure 3) itself (RDV resistance assays) is the evidence of acquiring RDV resistance via C799F mutation with regard to the early mutant viruses (CH-LT1m to CH-LT3m). The viral properties of CH-LT4 and later isolates were greatly changed not only with regard to fusogenicity but also in growth kinetics (please see the reply to the next comment). Due to this drastic change, it was very difficult to compare all the isolates under the unified condition (this was part of the reasons why we did not include CH-LT4 and later isolates in the **Figure 6** (RDV resistance assay). Alternately, we performed the comparison experiments of the later isolates with the original isolate CH-LT1 in the presence or absence of 5µM RDV at a fixed point (3 d.p.i for CH-LT1 and 4 d.p.i for CH-LT4 and later isolates). We have presented this result as a **Supplementary Figure S2**. As shown in the figure, CH-LT4 and later isolates could propagate at least to some extent (approximately at one to two order orders of magnitude) in the presence of 5µM RDV, whereas CH-LT1 was completely suppressed to undetectable levels. These results were described in the revised manuscript, as highlighted (Section of ‘Acquisition of RDV resistance via nsp12 mutations’ in Results)

Comment 5

The authors performed the RDV resistance assay with three isolates (CH-LT1m, CH-LT2m, and CH-LT3m), showing slightly high EC₅₀ values (1.75-fold, 3.38-fold and 2.38-fold increases, respectively). Although the other isolate CH-LT4m has the same C799F mutation as CH-LT2m that could confer RDV resistance, CH-LT4m was not included in this assay. Why not? It should be included.

Response to Comment 5

We thank the reviewer for this comment. The reasons why we did not include CH-LT4 and other later isolates in the RDV resistance assay are as follows:

- 1) As shown in the growth-kinetics (new **Figure 5**) which we added in the revised manuscript, the growth kinetics profiles of CH-LT4 and later isolates were quite different from those of earlier isolates. The earlier ones propagate rapidly and reach the peak at 2 to 3 d.p.i. Meanwhile, the later

ones propagate relatively slowly and reach a peak at 4 d.p.i. even though the MOI were set at the same value. Besides, the later isolates became fusogenic, as shown in **Figure 4**. It was difficult to determine resistance assay timing due to these non-negligible changes in viral properties.

- 2) The later isolates possess a great variety of mutations not only in the Spike protein but also in other genomic regions compared to CH-LT1 (BA.5.2.1). Our interest was mainly the role(s) of mutations found in nsp12 which appeared to be responsible for conferring the RDV resistance. For these reasons, we focused on the earlier isolates.

The RDV resistance of CH-LT4 and later isolates is shown in **Supplementary Figure S2**. The later isolates could grow even in the presence of 5 μ M RDV whereas CH-LT1 exhibited no responsiveness against RDV.

Comment 6

Statistical analysis was not provided for the RDV resistance assay.

Response to Comment 6

We thank the reviewer for this comment. We completely agree. In fact, we performed the RDV resistance assay in triplicate, but only plotted the mean values simply to avoid the complexity of the graph in the previous manuscript. We prepared the new modified **Figure 6**. The 95% confidence interval was shown in Figure 6 and the Results as follows ‘The EC₅₀ of CH-LT1m, CH-LT2m, and CH-LT3m was 1.4 (95% CI: 1.1–1.7 μ M), 2.7 (95% CI: 2.0–3.3 μ M), and 1.9 μ M (95% CI: 1.4–2.5 μ M), respectively.’ These changes are highlighted and found in the revised manuscript. (Section of ‘Acquisition of RDV resistance via nsp12 mutations’ in Results).

Comment 7

To validate the RDV resistance of the isolates, growth kinetics in the presence and absence of RDV treatment should be provided.

Response to Comment 7

Thank you for this comment. As requested, we performed the growth kinetics experiments for all the isolates in triplicate, and the results are shown in the new **Figure 5**. The difference in growth kinetics between the earlier isolates and the later isolates can be seen clearly in this Figure. Since we added the new data, the manuscript was modified accordingly in the Results, Discussion, and Methods, as highlighted. If we include the data in the presence of RDV, the graph would be much more congested. RDV resistance in the earlier isolates was already shown in **Figure 6**. Therefore, we performed the

comparison of later isolates and the original strain (CH-LT1) in the presence or absence of 5 μ M RDV at a fixed point as described in our response to your comment 4. We provide these data as a **Supplementary Figure S2**.

Comment 8

IgG antibody titers against SARS-CoV-2 S and N, described in lines 63–65, should be provided along with the Ct value in Table 1.

Response to Comment 8

We thank the reviewer for this comment. We modified **Table 1**.

Comment 9

In Table 1, the Ct value on Day 78 is “2.4”. This must be incorrect.

Response to Comment 9

We thank the reviewer for this comment. The correct value is “28.4.” Corrected figures are shown in **Figure 1** and **Table 1**.

Comment 10

In lines 86 and 89, “20 (11+9) locations” and “50 (21+29) locations”. An explanation of what the numbers in parentheses indicate is needed.

Response to Comment 10

We thank the reviewer for this comment. We modified as follows “ORF1ab harboured the highest number of mutations, identified at 50 (ORF1a and ORF1b harboured 21 and 29, respectively) loci, followed by the S protein, with mutations in seven loci.” These changes are highlighted and found in the revised manuscript (Section of ‘SARS-CoV-2 genetic mutations’ in Results).

REVIEWER COMMENTS

Reviewer #2 (Remarks to the Author):

I appreciate the author's revisions. They have clarified the clinical case to my satisfaction, and I have no further comments in this regard.

Reviewer #3 (Remarks to the Author):

Most of my concerns have been addressed, but a few minor issues remain.

To authors' response to comment 1:

- I pointed out that the statement in lines 109–111, “The amino acid mutations compared with those in the confirmed BA.5.2.1 genome (same strain as in this case) were categorised as major and minor (by a frequency of more and less than 50%, respectively),” should be changed to, “The amino acid mutations compared with those in the Wuhan genome”. This is a necessary change because, for example, it states in lines 112–113 that, “Fifty-four major mutations were identified on Day 22, and the number increased to 70 by Day 86.”; however, these numbers were the ones when compared to the Wuhan stain. Otherwise, accurate major and minor mutation numbers compared to the BA.5.2.1 should be provided in this sentence.

- Was the phylogenetic tree in new Figure 3 generated based on the whole genome sequence, NSP12, or the spike sequence? This information should be provided.

To authors' response to comment 3:

- In new Figure 4a-d; it is unclear which images represent which isolates.

Reviewer #4 (Remarks to the Author):

The manuscript tackles an important problem and, as noticed by reviewer 1, underscores the complexities of managing COVID-19 in immunocompromised patients.

While most of the previous critiques were adequately addressed, the authors failed to properly take into account that the mutational patterns observed in their patient, who received an early vaccine targeting the Spike protein of the initial wild-type strain, is not unusual. They did modify their sentence to acknowledge this; however, they failed to explain how their overall findings provide, at this point, any significantly new insight.

Moreover, the reply to comment of reviewer 1, asking for a phylogenetic analysis is wholly inadequate. Specifically:

1. No details are given about the tree that was inferred
2. Scaled branch lengths in nucleotide differences suggest the authors did not use a substitution model to infer genetic distances, thus failing to correctly estimate the total number of mutations.
3. No bootstrap values are provided nor any other statistical measures to assess how supported the tree branching pattern is, making impossible to evaluate the reliability of the tree.
4. Last but not least, the simple comparison of the patient's sequences with a single strain of BA.5.2.1 is meaningless. How and why was that BA.5.2.1 strain chosen? Without a Pango lineage classification or a more thorough phylogenetic analysis including multiple omicron strains, how can they be sure of any specific sub-variant their strains belong to?

Point-to-Point Responses to the Reviewers

Manuscript ID: NCOMMS-23-23503

Reviewer #2 (Remarks to the Author):

I appreciate the author's revisions. They have clarified the clinical case to my satisfaction, and I have no further comments in this regard.

Response to Reviewer #2

We thank the reviewer for this comment. We have prepared a revised manuscript in consideration of the comments from other reviewers.

Point-to-Point Response to the Reviewers

Manuscript ID; NCOMMS-23-23503

Reviewer #3 (Remarks to the Author):

Most of my concerns have been addressed, but a few minor issues remain.

Response to Reviewer #3

We thank the reviewer for these very positive comments on the work. We have provided point-to-point responses to every question raised.

To authors' response to comment 1:

- I pointed out that the statement in lines 109–111, “The amino acid mutations compared with those in the confirmed BA.5.2.1 genome (same strain as in this case) were categorised as major and minor (by a frequency of more and less than 50%, respectively),” should be changed to, “The amino acid mutations compared with those in the Wuhan genome”. This is a necessary change because, for example, it states in lines 112–113 that, “Fifty-four major mutations were identified on Day 22, and the number increased to 70 by Day 86.”; however, these numbers were the ones when compared to the Wuhan stain. Otherwise, accurate major and minor mutation numbers compared to the BA.5.2.1 should be provided in this sentence.

- Was the phylogenetic tree in new Figure 3 generated based on the whole genome sequence, NSP12, or the spike sequence? This information should be provided.

Response to “To authors' response to comment 1”

We thank the reviewer for this comment.

We have revised the text based on your feedback as follows: “The amino acid mutations compared with those in the **index strain from Wuhan**”. We also modified the legend of Fig. S1.

The phylogenetic analysis was performed using the whole genome sequence. This information is described in Methods section.

To authors' response to comment 3:

- In new Figure 4a-d; it is unclear which images represent which isolates.

Response to “To authors' response to comment 3”

We thank the reviewer for this comment.

Accordingly, we have modified the legend of Figure 4 as follows:

a: CH-LT1, isolated on Day 22, b: CH-LT1m, another variant with a V792I mutant in nsp12, isolated on Day 22, c: CH-LT2m, a variant with a C799F mutant in nsp12, isolated on Day 29, d: CH-LT3m, a variant with a M794I mutant in nsp12, isolated on Day 36, e: CH-LT4, a variant with C799F mutant in nsp12, isolated on Day 62, f, g: CH-LT6 and 7, variants with C799F mutant in nsp12, isolated on Day 78 and 83, respectively, h: control.

Point-to-Point Response to the Reviewers

Responses to Reviewer Comments

Manuscript ID; NCOMMS-23-23503

Reviewer #4 (Remarks to the Author):

The manuscript tackles an important problem and, as noticed by reviewer 1, underscores the complexities of managing COVID-19 in immunocompromised patients.

While most of the previous critiques were adequately addressed, the authors failed to properly take into account that the mutational patterns observed in their patient, who received an early vaccine targeting the Spike protein of the initial wild-type strain, is not unusual. They did modify their sentence to acknowledge this; however, they failed to explain how their overall findings provide, at this point, any significantly new insight.

Moreover, the reply to comment of reviewer 1, asking for a phylogenetic analysis is wholly inadequate. Specifically:

Response to Reviewer #4

We thank the reviewer for these very positive comments on the work. We have provided point-to-point responses to every question raised.

Regarding the first half of the comment “They did modify their sentence to acknowledge this; however, they failed to explain how their overall findings provide, at this point, any significantly new insight”, we have addressed this issue by further elaborating the text in the Discussion as follows:

“The patient received the initial two doses of the SARS-CoV-2 mRNA vaccine and an additional booster dose, all based on the index virus from Wuhan. The increase in neutralizing antibody titres against the Omicron BA.5 subvariant was lower compared to the response against the index virus from Wuhan²⁴. In our case, the BA.5 subvariant may have evaded neutralizing antibodies induced by these vaccinations. Considering the emergence of such mutant strains, the World Health Organization addressed the need for vaccine composition updates, including the Omicron strain, in June 2022²⁵. Reflecting on our case, the IgG titres against the S protein, measured at 142-190 U/mL after Day 75, were considerably lower than those observed in healthcare workers²⁶. As such, the neutralizing antibody titre against the BA.5 subvariant was expected to be limited even after BA.5 infection. Considering both the attenuated immune condition post lung transplantation and the immunological escape of the BA.5 subvariant, these factors may have contributed to the severe clinical course observed in our patient.”

24. Hachmann, NP. et al. Neutralization escape by SARS-CoV-2 Omicron subvariants BA.2.12.1, BA.4, and BA.5. *N Engl J Med.* **387**, 86-88 (2022).
25. World Health Organization. Interim statement on the composition of current COVID-19 vaccines - 17 June 2022. <https://www.who.int/news/item/17-06-2022-interim-statement-on--the-composition-of-current-COVID-19-vaccines> (2023).
26. Igari, H. et al. Antibody responses and SARS-CoV-2 infection after BNT162b2 mRNA booster vaccination among healthcare workers in Japan. *J Infect Chemother.* **28**,1483-1488 (2022).

Comment 1

No details are given about the tree was inferred

Response to Comment 1

We thank the reviewer for this comment.

The details of phylogenetic analysis are now shown in the Methods section.

Comment 2

Scaled branch lengths in nucleotide differences suggest the authors did not use a substitution model to infer genetic distances, thus failing to correctly estimate the total number of mutations.

Response to Comment 2

We thank the reviewer for this comment.

The phylogenetic tree was re-analysed by using MAFFT and IQ-TREE. The current strains and specific VOC were compared and shown in the revised Figure 3b. The best-fit substitution model was selected by ModelFinder plus option of IQ-TREE.

Details on the software used for these analyses have been added to the "Data availability" section.

Comment 3

No bootstrap values are provided nor any other statistical measures to assess how supported the tree branching pattern is, making impossible to evaluate the reliability of the tree.

Response to Comment 3

We thank the reviewer for this comment.

The bootstrap resampling was performed, and the values are described in the phylogenetic tree in the revised Figure 3b.

Comment 4

Last but not least, the simple comparison of the patient's sequences with a single strain of BA.5.2.1 is meaningless. How and why was that BA.5.2.1 strain chosen? Without a Pango lineage classification or a more thorough phylogenetic analysis including multiple omicron strains, how can they author be sure of any specific sub variant their strains belong to?

Response to Comment 4

We thank the reviewer for this comment.

The determination of the lineage was performed by using Pangolin and Nextclade web applications. The Nextclade-based phylogenetic placements of current strains in all SASR-CoV-2 lineages are shown in the revised Figure 3a.

REVIEWERS' COMMENTS

Reviewer #4 (Remarks to the Author):

I found the revised version of the manuscript has thoroughly addressed any previous concern. I have no further comments.